# AgentSuite: Toward More Reliable Agent Evaluation with a Component-Based Benchmark Auditing Pipeline

**Hyewon Suh** [* 1] **Binfei Ji** [* 1] **Seojune Lee** [* 1] **Rishi Khare** [1] **Basit Khan** [1] **Hyunjun Kim** [1] **Tianyi Zhang** [1]
**Venkat Krishna Srinivasan** [2] **Peter Belcak** [2] **Shizhe Diao** [2] **Pavlo Molchanov** [2] **Yingyan (Celine) Lin** [1]
**Zhen Dong** [2 3]

## Abstract

Reliable evaluation of large language model (LLM) agents depends critically on benchmark validity, yet modern agent benchmarks often contain hidden flaws arising from interactions among user instructions, environments, tools, ground-truth trajectories, and evaluation protocols. These flaws confound model errors with benchmark artifacts and undermine leaderboard-based comparisons. We propose **COBA** (**CO**mponent-based **B**enchmark **A**uditing), an automated pipeline for diagnosing and filtering validity issues in agent benchmarks. COBA decomposes agent tasks into four standardized components—User, Environment, Ground Truth, and Evaluation—and operationalizes a component-level issue taxonomy using hybrid rule-based detectors and taxonomy-guided LLM evaluation. Across six widely used agent benchmarks, COBA achieves strong alignment with expert judgments, with F1 scores between 0.791 and 0.874. It complements manual verification of $\tau^2$-bench by identifying issues missed due to benchmark complexity, and generalizes to previously unseen benchmarks with minimal adaptation. Our analysis shows that benchmark flaws are widespread and materially affect evaluation outcomes, demonstrating that component-based auditing provides a scalable foundation for more reliable and interpretable agent evaluation. We release AgentSuite, a unified benchmark-running platform that includes the COBA auditing pipeline and audited benchmark variants: https://github.com/Agent-Suite/AgentSuite.

[1]Georgia Institute of Technology [2]NVIDIA [3]University of California, Santa Barbara. Correspondence to: Zhen Dong <zhend@nvidia.com, zhend@ucsb.edu>.

*Proceedings of the $43^{rd}$ International Conference on Machine Learning*, Seoul, South Korea. PMLR 306, 2026. Copyright 2026 by the author(s).

## 1. Introduction

Research on agentic systems has rapidly expanded to enhance the practical utility of large language models (LLMs) (Park et al., 2023), leading to the development of diverse agents across domains such as tool use, web navigation, and simulated environments. As agent capabilities advance, benchmark-driven evaluation has become central to measuring progress (Zheng et al., 2023; Liu et al., 2024). However, the reliability of agent benchmarks remains underexamined, raising concerns about the validity of model comparisons and leaderboard-based conclusions.

Prior work on benchmark quality has largely focused on static or task-specific datasets, relying on manual verification, domain-specific checks, or limited automated heuristics (Northcutt et al., 2021; Truong et al., 2026; Vendrow et al., 2025). While these approaches have revealed pervasive issues in conventional benchmarks, they face substantial challenges when applied to agent benchmarks. Agent benchmarks are inherently complex: tasks involve structured interactions among users, environments, tools, ground-truth specifications, and evaluation protocols. As a result, manual inspection does not scale, and existing automated methods are not designed to systematically capture semantic and contextual issues arising from interactions among benchmark components (Truong et al., 2026; Zhu et al., 2026). A comprehensive discussion of related work is provided in Appendix A.

To address these challenges, we propose the **COBA** (**CO**mponent-based **B**enchmark **A**uditing) pipeline, a component-based automated auditing pipeline for agent benchmarks. The COBA pipeline systematically diagnoses benchmark issues by combining rule-based detectors for deterministic errors with structured LLM-based validation for semantic and contextual flaws. As an optional refinement, we include a rebuttal step that reviews initially flagged cases to reduce false positives and facilitate human verification.

Our approach is grounded in a standardized, component-based representation of agent benchmarks. We define a hierarchical unit structure that decomposes agent benchmarks into four primary components—*User*, *Environment*,

*Ground Truth*, and *Evaluation*—along with their required sub-elements. This structure is derived through an analysis of six representative agent benchmarks spanning multiple domains, drawing inspiration from divide-and-conquer principles (Khot et al., 2023) and staged reasoning strategies such as least-to-most prompting (Zhou et al., 2023). Based on this representation, we construct a component-level issue taxonomy that captures recurring validity failures observed across benchmarks. This taxonomy serves as the conceptual foundation for automated diagnosis and directly informs the design of the COBA pipeline.

We evaluate the COBA pipeline on six widely used agent benchmarks and demonstrate strong alignment with expert judgments, achieving F1 scores between 0.791 and 0.874. Direct comparison with the manual verification process of $\tau^2$-bench (Cuadron et al., 2025) reveals that our pipeline complemented the shortcomings of manual inspection, demonstrating the efficacy of automation by identifying issues missed due to the inherent complexity of inspecting agent benchmarks. Furthermore, the pipeline generalizes effectively to previously unseen benchmarks, including BFCL V4 (Patil et al., 2025) and NexusBench (Nexusflow.ai team, 2024), with minimal adaptation.

Applying the COBA pipeline reveals that benchmark issues are widespread, affecting on average 27% of tasks. We further analyze how different issue types influence agent evaluation outcomes and show that removing flawed tasks leads to substantial changes in model leaderboard rankings, altering 60% of the model rankings in ACEBench. These results demonstrate that benchmark quality has a decisive impact on agent evaluation and model comparison.

In summary, our contributions are threefold: (1) we introduce a component-based issue taxonomy for diagnosing validity issues in agent benchmarks; (2) we propose the COBA pipeline, an automated auditing pipeline that achieves strong agreement with human experts and generalizes across benchmarks; and (3) we provide a large-scale empirical analysis of benchmark flaws and their impact on agent evaluation, and release AgentSuite, a unified benchmark-running platform that includes the COBA auditing pipeline and issue-guided fixes, supporting more reliable future benchmarks.

## 2. Systematic Analysis of Agent Benchmark Issues

### 2.1. Overview of Agent Benchmarks

We analyze six widely used LLM-agent benchmarks that span diverse settings and evaluation styles: *BFCL V3* (Patil et al., 2025), *ACEBench* (Chen et al., 2025), *Drafter-Bench* (Li et al., 2025), *$\tau$-Bench* (Yao et al., 2025), *$\tau^2$-Bench* (Barres et al., 2025), and *CFB* (Zhong et al., 2025). They differ along four structural components that together

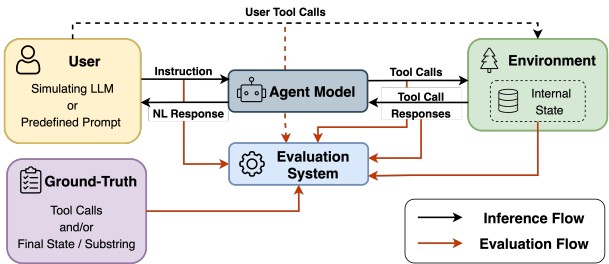

*Figure 1.* Illustrating generalized components of agentic AI benchmarks and their interactions.

define an agent benchmark setting (see Figure 1): **User** (e.g., predefined vs. LLM-simulated, single vs. multi-turn), **Environment** (e.g., stateless vs. stateful; tool/API availability and semantics), **Evaluation** (e.g., final-state checks, tool-call matching, LLM-based judging), and **Ground Truth** (e.g., full trajectories vs. milestone steps; policy constraints). This four-component decomposition exposes issues that static QA does not encounter and motivates a systematic taxonomy.

### 2.2. Component-Aligned Issue Taxonomy

We characterize and categorize recurrent benchmark issues by the component in which they originate, as shown in Table 2. Such a component-wise view turns scattered anecdotes into actionable categories and directly informs the modular detectors used in our pipeline (see Sec. 3).

*Table 1.* A concrete example: issue breakdown for $\tau$-Bench.

| Component | Share (%) |
|---|---|
| User | 21.0 |
| Environment | 30.6 |
| Evaluation System | 22.6 |
| Ground Truth | 25.8 |

The taxonomy is derived through a bottom-up annotation process. Five annotators with experience in LLM and agent-system research reviewed sampled tasks and trajectories. For each benchmark, three annotators independently proposed recurring issue types without a predefined schema. We iteratively refined the categories by comparing annotations, merging overlaps, splitting ambiguous cases, and resolving disagreements. Sampling was expanded until categories stabilized, covering 21–29% of tasks across the six benchmarks, which together contain 5,086 tasks. We do not claim the taxonomy is exhaustive, since a complete manual audit was not performed and future benchmarks may introduce new failure modes. Rather, it captures recurring issue patterns observed across current agent benchmarks and is designed to be extensible.

*Table 2.* A summary of the identified issue taxonomy of agent benchmarks.

| Benchmark Component | Issue Category | Description | Affected Benchmarks |
|---|---|---|---|
| **User** | **Ambiguous instruction** | The predefined user prompt is underspecified and allows multiple interpretations while the benchmark expects one specific task completion trajectory. | ACEBench, CFB |
| | **User role confusion** | The user simulator sends messages or behaves like an assistant rather than a user. | $\tau$-Bench, $\tau^2$-Bench |
| **Environment** | **Incorrect tool-call responses** | A tool returns inaccurate or irrelevant results that prevent the agent from completing the task correctly. | CFB, $\tau$-Bench, $\tau^2$-Bench |
| | **Insufficient toolset** | The environment does not provide the necessary tools for the agent to fulfill the user's request. | ACEBench, BFCL V3 |
| | **Misleading tool design** | Tool names or descriptions misrepresent their actual behavior. | ACEBench, $\tau$-Bench |
| | **Incorrect system prompt** | The system prompt itself contains errors or misleading examples that guide the agent toward invalid calls. | DrafterBench |
| **Evaluation System** | **Too lenient** | Evaluation criteria allow trivial or incomplete solutions to pass. | $\tau$-Bench |
| | **Too strict** | Evaluation criteria unfairly penalize semantically correct answers for minor deviations. | ACEBench, CFB |
| **Ground Truth** | **Malformed tool calls** | Ground-truth calls violate the function schema by using wrong types, invalid values, or missing arguments. | ACEBench, BFCL V3, CFB |
| | **Incorrect tool calls** | Ground-truth calls select the wrong function or parameters, contradicting the user's request or context. | ACEBench, CFB, BFCL V3, $\tau$-Bench, $\tau^2$-Bench |
| | **Redundant/ungrounded tool calls** | Ground-truth call sequences contain tool calls that are unnecessary or ungrounded by the context, causing unfair evaluation. | CFB |

**User-related issues.** User-side issues often stem from underspecified prompts that force agents to produce a single "correct" response despite open-ended instructions (e.g., some ACEBench and CFB tasks). In settings with *LLM-simulated users* (e.g., $\tau$-Bench and $\tau^2$-Bench) (Yao et al., 2025; Barres et al., 2025), we observe *role confusion* where the user model produces assistant-like confirmations (e.g., "your reservation has been canceled"), corrupting dialogue flow and making agent behavior hard to judge.

**Environment-related issues.** These arise when the actions available to the agent or the feedback it receives are inaccurate, misleading, or insufficient. (i) *Incorrect tool-call responses* provide wrong or irrelevant results even for correct queries, blocking task completion (e.g., CFB, $\tau$-Bench, and $\tau^2$-Bench). (ii) *Insufficient toolsets* omit necessary tools, rendering tasks unsolvable by construction (e.g., ACEBench and BFCL V3). (iii) *Misleading tool design* (names/descriptions that contradict actual behavior) steers agents toward suboptimal functions (e.g., ACEBench and $\tau$-Bench). (iv) *Incorrect system prompts* can hardwire invalid behavior—for example, a prompt instructing agents to call Python methods without parentheses (DrafterBench) produces systematically invalid calls.

**Evaluation-System Issues.** Evaluation criteria can be miscalibrated. Overly lenient scoring allows agents to exploit loopholes — for example, about 38% of $\tau$-Bench tasks pass if the database remains unchanged, enabling a "do nothing" strategy (Zhu et al., 2026). Conversely, overly strict criteria can reject semantically correct outputs due to brittle

exact-match requirements.

**Ground-truth issues.** Errors in the benchmark's *answer key* are especially harmful because they redefine correctness. We observe: (i) *Malformed tool calls* in the reference trajectories that violate schemas (wrong types/enums, missing required arguments), penalizing agents that adhere to the API. (ii) *Incorrect function or parameters* in ground truth that contradict user intent or policy (canceling a non-cancelable item), forcing agents to mimic mistakes to receive credit. (iii) *Redundant or ungrounded steps* that add unnecessary actions; efficient solutions are marked wrong for not reproducing superfluous calls.

**Discussion.** The aforementioned issues arise across all four components rather than being concentrated in one place. For example, in $\tau$-Bench (see Table 1), the shares are: User - 21.0%, Environment - 30.6%, Evaluation - 22.6%, and Ground Truth - 25.8%. This spread motivates a component-wise design of detectors; in Sec. 3, we operationalize the taxonomy into modular rules and LLM-judge checks in our COBA pipeline.

## 3. Methodology: From Taxonomy to Automated Filtering

We operationalize the component-based issue taxonomy into the **COBA pipeline**, an automated pipeline for detecting flaws in agent benchmarks at scale as illustrated in Figure 2. Our key observation is that, despite surface-level diversity, agent benchmarks share a common structural organization.

*Figure 2.* Overview of the end-to-end process of utilizing issue taxonomy, COBA pipeline, and the final cleaned benchmarks.

By mapping benchmark-specific formats into a standardized component representation, we enable a unified detection procedure across heterogeneous benchmarks.

The pipeline consists of two primary stages: (1) *Deterministic Structural Filtering*, which identifies explicit schema and role violations; and (2) *Taxonomy-Guided LLM Evaluation*, where a language model reasons about semantic and contextual flaws conditioned on the component definitions. We also include an optional *adversarial rebuttal* stage as an additional refinement to reduce over-flagging and to facilitate efficient human verification.

### 3.1. Problem Formulation: Standardized Task Representation

Agent benchmarks vary widely in domain, interaction protocol, and evaluation design. Some rely on fixed user prompts, others employ LLM-simulated users; environments may be stateful or stateless; evaluation ranges from exact tool-call matching to semantic outcome checks. To support automated analysis across this diversity, we define a **standardized agent task** representation that aligns with our component-based taxonomy:

$$T = \langle \mathcal{U}, \mathcal{E}, \mathcal{G}, \mathcal{V} \rangle. \tag{1}$$

Here, $\mathcal{U}$ denotes the **User** component, including the user instruction and, when applicable, user simulator specifications. $\mathcal{E}$ denotes the **Environment**, consisting of system policies, the initial state, and the available toolset with schemas and descriptions. $\mathcal{G}$ denotes the **Ground Truth**, represented as a reference action observation trajectory. $\mathcal{V}$ denotes the **Evaluation System**, including success criteria and matching rules.

**Purpose and Benchmark Mapping.** The standardized task representation enables a unified treatment of heterogeneous agent benchmarks. By explicitly separating user specifications, environment definitions, ground-truth trajectories, and evaluation criteria, it operationalizes the component-based taxonomy and localizes where issues originate. For each benchmark, we implement a lightweight mapping: a simple data-loading step that extracts native benchmark fields, such as instructions, tool schemas, ground truths, and evaluation rules, into $T$. This mapping is the only benchmark-dependent step. Once constructed, the same detection logic and prompt structure apply across benchmarks, with only minor benchmark-specific context, such as evaluation rules or design intent, included when needed for correct interpretation.

**Component-Wise Decomposition.** Rather than assessing task validity holistically, we adopt a component-wise verification strategy inspired by structured decomposition principles (Zhou et al., 2023). While the LLM judge is provided with the full task context, issue definitions are organized by component, guiding the model to reason about failures in a localized and systematic manner. This design is motivated by our empirical observation that benchmark issues are distributed across all components rather than concentrated in a single source.

Our objective is to infer a structured judgment function:

$$\phi(T) \to (y, c, r), \tag{2}$$

where $y \in \{0, 1\}$ indicates whether the task is flawed, $c$ denotes the corresponding issue category from the taxonomy, and $r$ is a natural-language rationale localizing the issue.

### 3.2. Stage 1: Deterministic Structural Filtering

Certain issue categories arise from explicit structural violations that can be detected without semantic reasoning. Examples include schema violations and role confusions in dialogue logs. We apply a set of rule-based detectors $\mathcal{H}(T)$ as an efficient first pass, removing structurally invalid tasks before invoking LLM-based evaluation. This stage prioritizes precision and computational efficiency and is not intended to capture higher-level semantic flaws.

### 3.3. Stage 2: Taxonomy-Guided LLM Evaluation

Many benchmark issues involve semantic inconsistencies across components, such as ambiguous user instructions, logically incorrect ground-truth actions, or mismatches between tool descriptions and observed behavior. Detecting such flaws requires reasoning over interactions among $\mathcal{U}$,

$\mathcal{E}$, $\mathcal{G}$, and $\mathcal{V}$. We therefore employ a language model as a judge $\mathcal{J}$, explicitly conditioned on the standardized task representation and the issue taxonomy.

**Judge Prompt Design.** The judge prompt follows a structured template comprising: (i) the complete standardized task $T$; (ii) the component-based issue taxonomy; and (iii) instructions specifying the required judgment and output format. This design provides full contextual grounding while constraining the model to reason within a fixed ontology, reducing free-form interpretation.

### 3.4. Adversarial Rebuttal for False-Positive Reduction

LLM-based evaluators can exhibit systematic alignment-related biases, including sycophantic tendencies that favor responses aligned with user-provided framing over objective correctness (Sharma et al., 2024). In our setting, we observe that judges instructed to identify issues may over-flag valid but unconventional solutions, leading to false positives.

To mitigate this effect, we introduce an optional **adversarial rebuttal** refinement for tasks initially flagged as flawed. A secondary agent $\mathcal{A}$ reviews tasks flagged as flawed and evaluates whether the judgment should be overturned under a small set of adjudication principles, such as allowance for valid workarounds, derivable parameters, or semantic equivalence when permitted by the evaluation rules.

Formally, for tasks initially labeled as flawed, the final decision is

$$y_{\text{final}} = \begin{cases} 0 & \text{if } \mathcal{A} \text{ overturns the judgment,} \\ 1 & \text{otherwise.} \end{cases} \tag{3}$$

This stage serves as a refinement mechanism to reduce false positives and to streamline downstream human verification. The primary taxonomy-guided judge alone already achieves strong alignment with expert annotations, as shown in Sec. 4. We report the ablation of this optional refinement in Appendix D.3.

## 4. Experiments and Results

### 4.1. Experimental Setup

**Evaluation protocol.** We evaluate the effectiveness of the COBA pipeline by measuring alignment with expert human judgments on benchmark issues. Our evaluation focuses on the LLM-based judgment stage, which targets semantic and contextual flaws that cannot be detected deterministically. We adopt two complementary human validation protocols.

**Human annotation.** We employed five annotators, all graduate researchers with prior publications or research experience in LLMs and agent systems. Annotators were trained with benchmark-specific documentation, implementation

walkthroughs, and annotation guidelines covering solvability, ambiguity, ground-truth correctness, evaluation fairness, and cross-component consistency. For each benchmark, three annotators independently labeled each sampled task for validity and, when flawed, assigned an issue category using the taxonomy with representative examples. We observe strong inter-annotator agreement before adjudication, with an average pairwise Cohen's $\kappa$ of 0.824. Disagreements, mainly involving borderline cases such as acceptable ambiguity versus underspecification, were resolved through discussion grounded in benchmark specifications, design intent, and evaluation logic. Further details are provided in Appendix B.1.

*Balanced subset validation.* For each benchmark, human experts annotate a stratified subset of tasks (10% of the benchmark, with a minimum of 30 tasks), sampled to contain an equal proportion of issue and non-issue cases. This protocol enables reliable estimation of precision, recall, and F1 scores for issue detection under controlled class balance.

*Post-hoc validation.* After running the full pipeline, experts manually verify all tasks flagged as issues across the entire benchmark. This protocol measures false positives at scale and produces the final issue-cleaned benchmarks used in subsequent analyses.

Rule-based structural filtering identifies deterministic violations by construction and is therefore excluded from human alignment evaluation, which centers on the taxonomy-guided LLM judgment stage.

**Implementation details.** We use Gemini-2.5-Pro-Thinking (Google, 2025) as the default LLM judge due to its strong reasoning performance in structured evaluation settings. We additionally evaluate judge robustness using Claude-4-Opus-Thinking (Anthropic, 2025) and DeepSeek-V3.1-Thinking (DeepSeek, 2025) in Appendix D.4. The results show consistent alignment trends across judge models, suggesting that the pipeline's effectiveness is not specific to a single LLM judge. Prompt templates for issue detection and adversarial rebuttal are provided in Appendix D.1. Across experiments, we evaluate 30 LLMs, including both proprietary and open-source models, as summarized in Table 11. For reproducibility, we release AgentSuite, a unified benchmark-running platform that packages the COBA auditing module and curated benchmark artifacts.

### 4.2. Main Results: Pipeline Validation

#### 4.2.1. HUMAN ALIGNMENT OF THE PIPELINE

We first evaluate the alignment of the COBA pipeline with expert judgments using the balanced subset validation protocol. Table 3 reports precision, recall, and F1 scores across six representative agent benchmarks. We exclude Drafter-Bench from this analysis, as all of its identified issues are

*Table 3.* Human alignment metrics, post-hoc validation accuracy, and final manually verified issue rates across benchmarks.

| Benchmark | Human Alignment | | | Val. Acc | Issue Rate |
|---|---|---|---|---|---|
| | Prec | Rec | F1 | | |
| ACEBench | 0.865 | 0.882 | 0.874 | 84.0% | 11.8% (121/1023) |
| BFCL V3 | 0.865 | 0.800 | 0.831 | 72.1% | 16.1% (129/800) |
| CFB | 0.878 | 0.720 | 0.791 | 87.1% | 16.2% (162/1000) |
| $\tau$-Bench | 0.846 | 0.733 | 0.796 | 92.0% | 24.2% (40/165) |
| $\tau^2$-Bench | 0.857 | 0.800 | 0.828 | 81.3% | 24.1% (67/278) |
| DrafterBench[1] | - | - | - | - | 66.7% (1280/1920) |

*Notes.* [1] DrafterBench: issues are dominated by duplicated tasks and a system prompt error. They are identified deterministically.

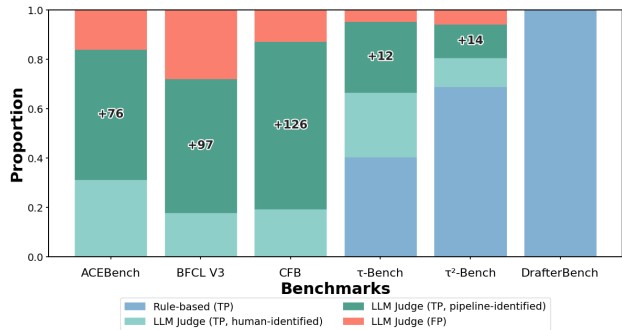

*Figure 3. Post-hoc* breakdown of tasks flagged as issues.

detected through deterministic structural filtering. The issue rates in Table 3 correspond to validated issues after post-hoc human verification of pipeline-flagged tasks, rather than raw LLM-judge outputs.

Across benchmarks, the pipeline achieves consistently strong agreement with expert annotations, with F1 scores ranging from 0.791 to 0.874. These results indicate that the taxonomy-guided LLM judgment stage reliably captures a broad range of benchmark issues identified by human experts.

We further analyze residual disagreements between COBA and human annotations in Appendix D.5. False positives primarily arise from overly strict reasoning under partial observability, while false negatives are dominated by missed low-level specification violations such as malformed tool calls. These concentrated error patterns suggest that remaining failures are predictable and could be addressed through targeted detector improvements.

To assess scalability beyond the sampled subsets, we conduct post-hoc validation on the full benchmarks. As shown in Table 3, the pipeline maintains high accuracy when applied at scale, with end-to-end validation accuracy exceeding 72.1% across benchmarks. Figure 3 summarizes the outcomes of manual verification for tasks flagged by the pipeline.

In addition to matching expert judgments, the pipeline surfaces a substantial number of previously undocumented issues. For example, in CFB, we identify up to 126 validated issues that were not previously reported. Together, these results demonstrate that COBA scales expert-level issue detection to large agent benchmarks while maintaining strong alignment with human judgment.

### 4.2.2. COMPARISON WITH MANUAL VERIFICATION OF $\tau^2$-BENCH

We further assess the practical utility of the COBA pipeline by comparing its outputs with the manually verified issues reported for $\tau^2$-Bench in concurrent work (Cuadron et al.,

2025). This comparison focuses specifically on *objective correctness issues*, namely violations that are independently verifiable and require correction for valid evaluation, and excludes clarity-, ambiguity-, or presentation-related annotations.

On the Airline and Retail domains of $\tau^2$-Bench, our pipeline recovers 11 out of the 15 objective issues identified through manual verification. In addition, it identifies 14 validated objective issues that fall within the same scope as the manual analysis but were not reported there.

Beyond these, our pipeline surfaces additional objective issue types that were outside the scope of the manual verification. Notably, we identify role-confusion issues that are prevalent in the Telecom domain, as well as "do-nothing" evaluation loopholes in the Airline and Retail domains. These issues arise from interactions between user simulation, environment design, and evaluation criteria, and require explicit component-level analysis to detect reliably.

Overall, this comparison suggests that taxonomy-guided automated auditing can complement manual verification by both increasing coverage of objective issues within existing scopes and surfacing additional objective issues in complex, multi-domain agent benchmarks.

### 4.3. Generalization to Unseen Benchmarks

We evaluate the generalization capability of the COBA pipeline by applying it to agent benchmarks that were not used during taxonomy construction or pipeline development. Specifically, we consider BFCL V4 (Patil et al., 2025) and NexusBench (Nexusflow.ai team, 2024), which differ substantially from the original benchmark set in domain coverage, task structure, and design. For each benchmark, we implement the standardized task mapping described in Sec. 3, extracting the required user, environment, ground-truth, and evaluation components. Beyond this benchmark-specific loading step, we reuse the same taxonomy-guided prompt templates and detection logic without modification.

*Table 4.* Alignment and validation metrics for generalization on unseen benchmarks.

| Bench | Human Alignment | | | Acc | Issue Rate |
|---|---|---|---|---|---|
| | Prec | Rec | F1 | | |
| BFCL V4 | 0.889 | 0.800 | 0.842 | 57.9% | 4.3% (11/255) |
| NexusBench | 0.956 | 0.705 | 0.811 | 80.6% | 12.6% (141/1118) |

*Table 5.* Disagreement analysis between taxonomy-guided and generic prompts on ACEBench and CFB.

| Benchmark | Disagreements | Tax-Prompt Win Rate |
|---|---|---|
| ACEBench | 114 | 72.8% |
| ComplexFuncBench | 202 | 73.3% |

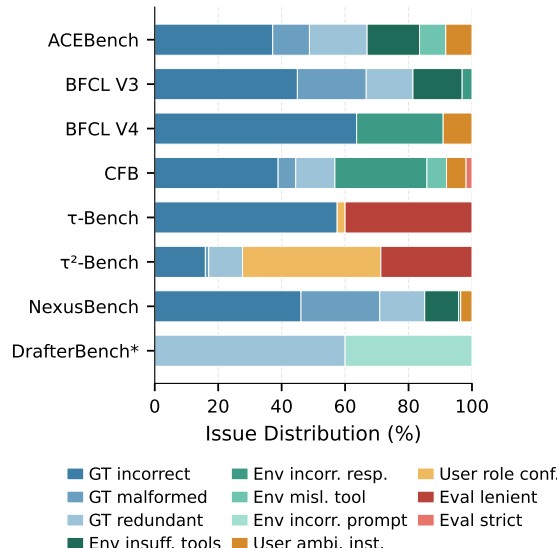

*Figure 4.* Issue distribution across benchmarks. Bars show the percentage breakdown by taxonomy category. Ground Truth issues (blues) dominate most benchmarks, while the $\tau$-Bench family shows distinct Evaluation (red) and User simulation issues (yellow).

Table 4 reports the human alignment results and issue statistics for both benchmarks. The pipeline achieves F1 scores of 0.842 on BFCL V4 and 0.811 on NexusBench, indicating strong agreement with expert judgments. In addition, COBA identifies 11 and 141 validated issues on BFCL V4 and NexusBench respectively, demonstrating that the issue taxonomy remains applicable to previously unseen benchmarks. These results suggest that the proposed pipeline and taxonomy generalize across benchmarks with different domains and interaction patterns, supporting their use as a reusable auditing framework for emerging agent benchmarks.

### 4.4. Ablation: Effect of Taxonomy-Guided Prompting

We evaluate the effect of the component-based issue taxonomy by comparing the full taxonomy-guided prompt with a generic critique prompt (Appendix D.2) that lacks structured component-based definitions.

To assess judgment quality, we focus on cases where the two prompts disagree. On ACEBench and CFB, there are 114 and 202 such disagreement cases respectively. We manually determine which judgment better reflects a valid benchmark issue. In these sampled disagreement sets, the taxonomy-guided prompt achieves win rates of 72.8% on ACEBench and 73.3% on CFB. The generic prompt also flags substantially more tasks as problematic, identifying $1.7\times$ more issues than the taxonomy-guided pipeline. These results indicate that while generic critique prompts tend to over-flag potential issues, taxonomy-guided prompting yields more reliable judgments when the two methods diverge, highlighting the role of structured, component-level guidance in constraining LLM-based evaluation. An ablation study of the adversarial rebuttal stage, evaluated as an additional refinement, is reported in Appendix D.3.

### 4.5. Benchmark Diagnosis and Design Insights

We analyze benchmark issues identified by the COBA pipeline to characterize recurring failure patterns and highlight design considerations that affect evaluation reliability, rather than to rank benchmark quality.

Figure 4 shows the component-wise distribution of issues across eight representative benchmarks. Issues consistently arise from all four components, motivating component-aware auditing. We summarize qualitative observations from three representative benchmarks below; additional analyses and full case studies of issues are provided in Appendix B.2.

**ACEBench.** ACEBench emphasizes realistic user behavior, including ambiguity in user intent. While this supports richer evaluation scenarios, many identified issues arise from underspecified instructions or mismatches between natural language requests and discrete parameter choices. This illustrates a trade-off between realism and evaluability, where benchmarks that permit intentional ambiguity require careful evaluation design to avoid penalizing reasonable agent behaviors.

$\tau^2$**-Bench.** $\tau^2$-Bench features complex multi-turn interactions, detailed policies, and LLM-simulated users. Beyond issues attributable to task complexity, we observe failures arising from interactions between user simulation and environment design. In particular, telecom tasks with dual control occasionally exhibit role confusion, where the user

simulator produces agent-like actions, revealing a subtle failure mode of user simulation in complex settings.

**BFCL (V3 and V4).** In BFCL, most issues are associated with ground-truth trajectories, including incorrect parameters, inconsistent units, or outdated reference information. Such errors often recur across tasks, indicating that small systematic issues can propagate widely. In BFCL V4, which includes real-time web search tasks, ground truths may additionally become stale as external information changes, highlighting a maintenance challenge for dynamic benchmarks.

Overall, these results suggest that many benchmark issues arise from increasing task complexity and realism rather than isolated implementation errors. By making such issues explicit, the proposed pipeline supports more informed benchmark design and usage without serving as a prescriptive quality score.

### 4.6. Impact of Benchmark Issues on Model Evaluation

We analyze how benchmark issues identified by COBA affect downstream model evaluation, focusing on ACEBench, where we conduct a detailed analysis of performance and ranking changes. Different issue types influence evaluation outcomes in qualitatively different ways, often with competing effects, making both absolute scores and relative rankings harder to interpret.

**Impact on model performance.** Based on their observed effects, benchmark issues can be grouped into four impact categories: (i) *inflation*, where lenient evaluation rewards unintended behaviors; (ii) *deflation*, where reasonable agent behavior is penalized; (iii) *unsolvable tasks*, which artificially cap achievable performance; and (iv) *noise*, which increases score variability without systematically shifting performance.

Figure 5 reports average model performance on ACEBench for tasks dominated by each category, alongside the cleaned benchmark. Inflation-prone tasks yield the highest apparent performance (1.000), while unsolvable tasks result in near-zero performance (0.035). Deflation-dominated (0.555) and noise-dominated tasks (0.666) both underperform the cleaned benchmark (0.780).

These results show that benchmark issues do not induce a consistent upward or downward bias. Instead, their combined presence introduces variance and obscures the relationship between measured performance and underlying agent capability, complicating the interpretation of aggregate scores.

**Impact on leaderboard rankings.** We further examine how issue removal affects relative model rankings on

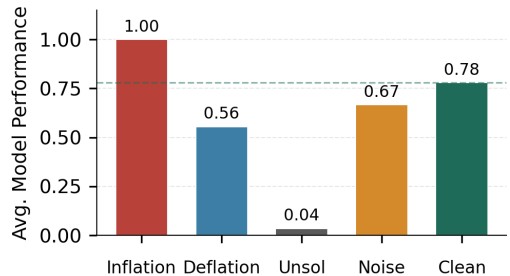

*Figure 5.* Model performance impact of issues in ACEBench

*Table 6.* Top 12 ACEBench Model Rankings: Initial vs. Cleaned benchmark. Dark green indicates rank improvement; dark red indicates a drop in rank. Full ranking is included in Appendix E

| Model | Initial Perf (Rank) | Cleaned Perf (Rank) |
|---|---|---|
| gpt-4o-20240806 | 85.2 (1) | 88.3 (1) |
| gpt-4.1 | 84.2 (2) | 87.2 (2) |
| gpt-4.1-mini | 82.0 (3) | 84.7 (4) |
| DeepSeek-V3.2-Exp-thinking-off | 81.6 (4) | 84.5 (5) |
| claude-4-sonnet-thinking-off | 81.3 (5) | 84.5 (6) |
| claude-4.5-sonnet-thinking-on-10k | 81.1 (6) | 85.0 (3) |
| claude-4-opus-thinking-off | 81.0 (7) | 84.5 (7) |
| Kimi-K2-Instruct-0905 | 80.8 (8) | 83.9 (12) |
| Kimi-K2-Instruct | 80.8 (9) | 83.6 (13) |
| o3-high | 80.6 (10) | 84.1 (8) |
| gpt-oss-120b | 80.4 (11) | 83.1 (16) |
| Gemini-2.5-flash-thinking-off | 80.3 (12) | 84.0 (11) |

ACEBench. Table 6 compares the top-12 model rankings before and after filtering validated issues. While the top two positions remain unchanged, several models experience non-trivial rank shifts in both directions, reflecting the heterogeneous effects of benchmark issues.

Overall, this analysis highlights that benchmark issues can materially affect both performance estimates and comparative evaluation. By identifying and removing such issues, the COBA pipeline supports more stable and interpretable model assessment. The full leaderboard is provided in Appendix E.

Because Table 6 is based on a single evaluation run per model due to computational cost, we further compare the observed rank changes against a random-removal baseline. Removing the same number of tasks at random over 100 trials yields substantially smaller shifts than issue filtering: the cleaned benchmark has Kendall's $\tau = 0.871$ versus 0.960 under random removal (95% CI: [0.935, 0.981]), and a pairwise flip rate of 5.75% versus 1.50% (95% CI: [0.46, 2.65]). This suggests that the ranking changes reflect systematic effects of validated benchmark issues rather than random task removal.

## 5. Discussion and Limitations

**Filtering and issue-guided repair.** The main evaluation in this work uses COBA for identifying and filtering flawed benchmark tasks, providing a conservative setting

for reliable agent evaluation. Beyond filtering, the same component-level diagnoses can also guide targeted benchmark repair when the required correction is clear, such as clarifying underspecified instructions or correcting ground-truth parameters. We treat such repair as a follow-up use case: repairs are applied manually using COBA's reasoning traces as guidance, and released as curated artifacts in AgentSuite where applicable. We provide representative repair examples in Appendix C.1 and analyze retention and issue-guided repair in Appendix C.2.

**Limitations.** Several limitations remain. First, the taxonomy captures recurring issue patterns observed across current agent benchmarks, but it is not exhaustive; future benchmarks may introduce failure modes outside the current categories. Second, COBA relies on LLM judges for semantic and contextual flaws and can inherit judge-specific biases, although taxonomy-guided prompting, adversarial rebuttal, and human verification help mitigate this risk. We provide further failure analysis and judge-model ablations in Appendix D.5 and Appendix D.4. Finally, the cleaned and repaired artifacts reflect issues detected by COBA and manually verified or corrected by us; they should be viewed as maintained benchmark artifacts rather than a guarantee that all remaining tasks are flawless.

## 6. Conclusion

We investigated the reliability of agent benchmarks and showed that hidden validity issues are pervasive in modern agent evaluation, arising from complex interactions among user instructions, environments, tools, ground-truth trajectories, and evaluation protocols. To address the scalability and systematic limitations of existing approaches, we proposed the COBA pipeline, a component-based benchmark auditing pipeline that combines a structured issue taxonomy with hybrid rule-based and LLM-based validation, with an optional adversarial rebuttal stage. Across diverse agent benchmarks, our pipeline achieves strong alignment with expert judgments, generalizes to previously unseen benchmarks, and reveals that benchmark flaws materially affect agent evaluation outcomes, including measured performance and model rankings. By providing a scalable, component-aware auditing framework and releasing cleaned benchmark artifacts, this work supports more reliable, interpretable, and reproducible evaluation of LLM-based agents.

## Impact Statement

This paper presents work whose goal is to advance the field of machine learning. There are many potential societal consequences of our work, none of which we feel must be specifically highlighted here.

## Acknowledgements

This work was supported in part by the National Science Foundation (NSF) under Award Nos. 2403297, 2312758, 2434166, 2048183, and 2016727. This work was also supported in part by CoCoSys, one of seven centers in JUMP 2.0, a Semiconductor Research Corporation (SRC) program sponsored by DARPA.

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

# A. Related Work

**Benchmark validity and issue detection.**   Validity issues have been shown to be pervasive across AI benchmarks, often distorting evaluation outcomes (Northcutt et al., 2021; Truong et al., 2026). Manual audits have uncovered systematic errors in widely used datasets, including ground-truth flaws in MMLU (Gema et al., 2025), while broader analyses demonstrate that label noise and ambiguity undermine benchmark reliability, motivating carefully curated "platinum benchmarks" (Vendrow et al., 2025). Task-specific validation methods, such as unit-test augmentation in SWE-Bench (Yu et al., 2025) or execution-based matching in SQL benchmarks (Yu et al., 2018), rely on deterministic semantics and executable environments, limiting their applicability beyond narrow domains.

More general automated approaches show mixed reliability. LLM-assisted detection achieves high precision on simple or saturated benchmarks but degrades in complex settings (Truong et al., 2026). Moreover, LLM-based evaluators exhibit conformity and egocentric biases, often favoring outputs aligned with their own priors rather than objective correctness (Zheng et al., 2023; Wang et al., 2024), resulting in low recall and unstable annotations even with enhanced prompting (Gema et al., 2025).

**Agent benchmark validation.**   Agent benchmarks introduce additional challenges due to their multi-component structure, where validity depends on interactions among users, tools, environments, ground-truth specifications, and evaluation protocols. Recent benchmarks reflect this complexity across diverse settings, including AgentBench (Liu et al., 2024), ToolBench (Qin et al., 2024), and WebArena (Zhou et al., 2024). While design-level guidance such as the ABC checklist (Zhu et al., 2026) provides principles for constructing rigorous benchmarks, validation largely relies on manual audits. Recent efforts reveal that underspecified tasks and annotation flaws in $\tau$-bench (Yao et al., 2025) impose artificial performance ceilings, motivating manually revised benchmarks such as $\tau^2$-bench (Cuadron et al., 2025). These approaches are effective but require substantial expert effort and do not scale.

In contrast, our work enables automated, task-level diagnosis of agent benchmark issues by decomposing validity into component-level issues while explicitly mitigating LLM evaluator biases.

# B. Annotation and Taxonomy Details

### B.1. Human Annotation Protocol

**Annotation setup.**   We employed five annotators, all graduate researchers with prior publications or research experience in LLMs and agent systems. For each task, three annotators independently labeled samples before discussion. Annotators were given access to the relevant benchmark artifacts, including task specifications, implementation details such as environment simulation and evaluation code, and model trajectories, enabling end-to-end understanding for informed judgment. Annotators were onboarded through documentation and code walkthroughs to establish a shared understanding of each benchmark's design intent and evaluation logic.

**Annotation guidelines.**   Annotators labeled a task as flawed only when there was clear, observable evidence of a benchmark-intrinsic validity issue. Such issues include inconsistencies across user, environment, ground-truth, and evaluation components; missing or incorrect environment support; ground-truth errors or contradictions with the benchmark specification; misaligned evaluation criteria; or underspecified instructions that prevent well-defined evaluation. Annotators were instructed not to label a task as flawed solely because it is difficult, admits evaluation-permitted ambiguity, or allows alternative valid solutions consistent with the benchmark design.

To calibrate decision boundaries, annotators reviewed representative edge cases. For example, an ACEBench task is labeled flawed when the ground truth specifies an unsupported year such as "2023" without evidence in the user prompt, since this introduces ungrounded information. In contrast, a case where "June" maps to "2022-06" is treated as valid when the mapping follows the tool's predefined enum schema. Additional calibration examples include cross-component inconsistency (e.g., CFB task where the user requests a taxi in NYC but the ground truth searches in LA) and missing environment support (e.g.,BFCL V3 task requesting travel time when only a distance-estimation tool is available), which render tasks inconsistent or unsolvable.

**Annotation procedure and adjudication.**   Annotators first made a binary validity judgment independent of the taxonomy and provided a rationale grounded in the benchmark specification. For flawed tasks, annotators then assigned an issue category based on the component where the root cause originates. For example, a $\tau^2$-Bench task that attempts to cancel an

already departed flight is categorized as a Ground Truth issue, since the reference action contradicts the environment state and policy.

Before discussion, we observe strong agreement, with an average pairwise Cohen's $\kappa$ of 0.824. Disagreements primarily involved borderline cases, such as whether an underspecified instruction should be treated as acceptable ambiguity. For instance, annotators initially disagreed on an ACEBench task where the user requested "some climate data" but the tool schema required a specific detail-level parameter. The case was ultimately labeled flawed because selecting a specific detail level introduced unsupported information and prevented a stable mapping to a single expected trajectory. All disagreements were resolved through discussion grounded in task specifications and benchmark evaluation criteria.

### B.2. Benchmark Diagnosis and Case Studies

This appendix provides additional benchmark diagnosis and case studies of different issues that expand on Section 4.5 and Section 2.

**ACEBench.** We identify issues arising from user queries, provided toolset, and the ground truth annotation. First, some user queries are underspecified; for example, a task that asks for "some climate data" fails to specify the detail level, which is required by the tool schema. Second, the toolset can be insufficient to solve some problems and sometimes misleading: one example is `vlookup_formula_generator`, which has a parameter `exact_match` that performs an exact match when set to false. Finally, ground truth annotations are often flawed or contain malformed inputs.

**BFCL V3.** We find the issues in the available toolsets and the ground truths. First, the toolset is insufficient for certain tasks; for instance, a user requests a travel-time estimate, but only a distance-estimation tool is provided. Second, the ground truth contains various errors: malformed calls using strings instead of integers, flawed or redundant tool calls, such as using an incorrect file name or requiring unnecessary sorting before counting the characters in file system-handling tasks.

**CFB.** This benchmark contains issues regarding the integrity of the environment, evaluation system, and the ground truth. First, the environment yields incorrect tool responses, such as resolving "Melbourne" to Florida instead of Australia. The evaluation system is occasionally too strict, requiring exact string matches for coordinate values, marking tool calls that use rounded values wrong. Additionally, the ground truths are plagued by data type violation, incorrect parameter values (e.g., searching LA instead of requested NYC), and redundant steps.

**DrafterBench.** We find an error in the system prompt design. The prompt provided to the agent contains syntactically incorrect Python code examples, instructing to perform a method call without parentheses.

$\tau$**-Bench.** We observe problems in user simulation, tool descriptions, and evaluation validity. First, the user simulator exhibits role confusion, producing assistant-like messages such as "I can look up your reservation". Second, some tool definitions are misleading. An example is `search_onestop_flight` being described as searching for direct flights. Third, the evaluation is excessively lenient, allowing a trivial "do-nothing" model to achieve a 38% success rate. Lastly, the ground truth contains policy violations, such as canceling basic-economy reservations, which is explicitly prohibited by the given system policy.

$\tau^2$**-Bench.** This benchmark suffers from user role confusion, environment errors, and incorrect ground truth. Similar to $\tau$-bench, the user model occasionally experiences role confusion and the ground-truth actions often violate the given policy. Additionally, we discover that the tool responses are sometimes unreliable, such as retrieving orders for a user different from the requested ID.

**NexusBench.** This benchmark mainly suffers from flawed ground-truth trajectories, most commonly involving malformed or semantically invalid tool calls (e.g., parameter values that contradict tool schemas). Many tasks are therefore unsatisfiable by a correct agent without violating the tool interface.

**BFCL V4.** This benchmark includes ground truth issues where reference trajectories include factual errors or the ground truth labels contradict the provided question and context.

Table 7 summarizes all identified issues, organized by the benchmark component, their specific issue category, and representative examples drawn from each benchmark.

*Table 7.* Comprehensive summary of benchmark issues and representative cases

| Benchmark Component | Issue Category | Benchmarks & Representative Cases |
|---|---|---|
| **User** | **Ambiguous instruction** | **ACEBench**: A user asks for "some climate data," but the function requires a detail-level field (Summary/Detailed), making the request ambiguous. |
| | **User role confusion** | $\tau$**-Bench**: The user model says "No worries! I can look up your reservation using your user ID." 
 $\tau^2$**-Bench**: The user model responds "Your reservation has been canceled." |
| **Environment** | **Incorrect tool-call responses** | **CFB**: The call `Search_Flight_Location` resolves the input "Melbourne" to Florida instead of Australia. 
 $\tau^2$**-Bench**: A tool retrieves orders made by `sofia_hernandez_5364` instead of `sofia_hernandez_8513` as it is asked. |
| | **Insufficient toolset** | **ACEBench**: The sample forces the agent to use a culturally focused landscape tool to answer a broad land-use change request. 
 **BFCL V3**: The user requests a travel-time estimate, but only a distance-estimation tool is available. |
| | **Misleading tool design** | **ACEBench**: The parameter `exact_match` in `vlookup_formula_generator` contradicts its actual behavior by performing an exact match when it is set to false. 
 $\tau$**-Bench**: The function `search_onestop_flight` is described as searching direct flights, which is misleading. |
| | **Incorrect system prompt** | **DrafterBench**: The system prompt shows Python code that calls a method without parentheses. |
| **Evaluation System** | **Too lenient** | $\tau$**-Bench**: A trivial do-nothing model succeeds in 38% of cases. |
| | **Too strict** | **CFB**: The evaluation system requires an exact string match for latitude/longitude, despite multi-dimensional matching being allowed elsewhere. |
| **Ground Truth** | **Malformed tool calls** | **ACEBench**: The field `schedule.time` violates the required HH:MM regex because it provides a time range ("08:00–17:00") instead of a single time. 
 **BFCL V3**: A ground-truth call `close_ticket` is invoked with the string `ticket_id` instead of an integer. 
 **CFB**: A ground-truth call provides latitude/longitude as floats instead of strings, violating the schema. |
| | **Incorrect tool calls** | **ACEBench**: The ground truth schedules the task with a "High" priority instead of the requested "Urgent," using an inappropriate scheduling tool. 
 **BFCL V3**: A ground-truth call creates `note.md` instead of the requested `notes.md`. 
 **CFB**: A ground truth searches for a taxi in LA, although the request was for NYC. 
 $\tau$**-Bench**: A ground truth cancels a basic-economy reservation, violating policy. 
 $\tau^2$**-Bench**: A ground truth cancels a departed flight, violating policy. |
| | **Redundant/ungrounded tool calls** | **BFCL V3**: The agent is asked to display last ten lines after sorting a file; The ground truth calls `sort` followed by `tail`, while `tail` call, which prints last lines of the original, unsorted file, is redundant. 
 **CFB**: The user instructs the agent to continue until it finds an attraction that meets a specified criterion, but the ground truth invokes `Get_Attraction_Details` in an arbitrary order and continues after the condition is met, producing redundant tool calls. |

## C. Repairable Cases and Retention Analysis

### C.1. Repairable Cases Enabled by the Pipeline

While our evaluation focuses on filtering, the structured reasoning traces produced by the pipeline also naturally support task repair when benchmark maintainers prefer fixing over removal. Below we provide three representative examples where the reasoning trace directly identifies the minimal edit needed to correct a flawed task.

#### C.1.1. BFCL: GROUND-TRUTH FILENAME ERROR (GROUND-TRUTH ISSUE).

**Task:** `multi_turn_long_context_10`

**Reasoning trace:** "Turn 3: The user requests creation of a file named `notes.md`, but the ground truth calls `touch(file_name='note.md')`, which misspells the filename. Turn 4 again refers to `notes.md`, while the ground truth continues with the incorrect name. This contradicts the user's instructions in both turns."

**Minimal fix:** Replace all occurrences of `note.md` with `notes.md` in the ground-truth sequence.

#### C.1.2. ACEBENCH: API SCHEMA VIOLATION (ENVIRONMENT ISSUE).

**Task:** `normal_single_turn_single_function_50`

**Reasoning trace:**

"The user wants to create a morning routine for their son. The ground-truth function call uses the

*Table 8.* Category-level retention before and after issue-guided fixes. Mean retention is reported with the minimum category retention in parentheses.

| Benchmark | Mean Retention (min) | Fixable (%) | Mean Retention+Fix (min) |
|---|---|---|---|
| ACEBench | 0.84 (0.27) | 80% | 0.98 (0.86) |
| CFB | 0.85 (0.81) | 20% | 0.87 (0.84) |
| $\tau$-bench | 0.67 (0.46) | 58% | 0.86 (0.78) |
| $\tau^2$-bench | 0.70 (0.44) | 39% | 0.81 (0.62) |
| BFCLv3 | 0.84 (0.81) | 73% | 0.96 (0.95) |
| DrafterBench | 0.33 (0.25) | 100% | 1.00 (1.00) |

`FamilyRoutineManager_createMorningRoutine` tool, which is correct for the task. However, the parameter `startTime` is set to '07:30'. The API schema restricts `startTime` to one of {'06:00', '07:00', '08:00'}, so the call violates the tool specification."

**Minimal fix:** Adjust the argument to a valid enum (e.g., '07:00') or update the schema to allow free-form times.

### C.1.3. COMPLEXFUNCBENCH: UNDERSPECIFIED USER INSTRUCTION (USER ISSUE).

**Task:** `Car-Rentals-49`

**Reasoning trace:** "The user asks for the booking summary of all vehicles that meet 'the conditions for reimbursement', but these conditions are never defined. The ground truth arbitrarily selects three vehicles to summarize, which is unsupported by the user's request and logically inconsistent."

**Minimal fix:** Specify the reimbursement conditions explicitly in the user prompt (e.g., price threshold, mileage, or rental duration).

These examples illustrate how the pipeline's component-level reasoning traces substantially reduce the effort needed to diagnose and repair flawed tasks, complementing the primary use case of issue filtering.

### C.2. Retention and Issue-Guided Repair Analysis

To assess whether filtering flawed tasks changes benchmark coverage, we measure category-level retention before and after issue filtering. Retention remains high for most benchmarks, typically at or above 0.8, while lower-retention cases are concentrated in benchmarks with systematic issues, such as user-simulation role confusion in $\tau$-Bench or duplicated/system-prompt issues in DrafterBench.

We further conduct a follow-up repair analysis in which task-level issues are manually corrected using COBA's taxonomy and component-level reasoning traces as guidance. After applying these targeted fixes, retention improves substantially across benchmarks, as shown in Table 8. These results indicate that many removed tasks are recoverable through localized edits, while remaining low-retention cases often reflect broader systematic benchmark flaws that require more extensive maintenance.

Overall, filtering removes concentrated invalidity while preserving most capability coverage, and issue-guided repair can recover many affected tasks. We release these curated artifacts as part of AgentSuite.

## D. Prompt Templates and Ablations

### D.1. Taxonomy-Guided Evaluation Prompt Template

To ensure transparency and reproducibility of our benchmark construction and filtering process, we include the exact evaluation prompt used for taxonomy-guided issue detection in CFB. This prompt is used to assess whether individual benchmark samples contain fundamental flaws that would invalidate their use in evaluation.

The evaluator is instructed to examine the provided benchmark component, and the prompt operationalizes a structured issue taxonomy that categorizes flaws according to their source, including user ambiguity, environment or tool inconsistencies,

and errors in the ground-truth solution itself.

Importantly, the prompt enforces a strict output schema and requires explicit categorization of detected issues. This design enables consistent automated aggregation of evaluator judgments and ensures that benchmark filtering decisions are traceable to well-defined error categories rather than subjective preferences.

---

**Taxonomy-guided LLM Evaluation Prompt for CFB**

```
You are an expert evaluator for ComplexFuncBench, a benchmark designed to assess the complex function-calling
    capabilities of LLMs.
Your task is to determine if a given benchmark sample has a fundamental flaw in its user prompt, environment,
    or ground-truths, which would make it unable to be incorporated in the evaluation.

You will be provided with:

* **User Prompt:** The user's task description and instructions.
* **Available Functions:** A complete list of functions the agent can use, including their JSON schemas.
* **Ground-Truth Trajectory:** The reference sequence of function calls. Note that messages with '"role":
    "observation"' are the outputs of the immediately preceding function call.

A sample is **flawed** if it exhibits one or more of the issues described below.

## Flaw Categories

Below is the categorization of benchmark issues, outlined according to its **relevant benchmark component**. A
    sample is considered flawed if it has one or more of the issues below.

### User

* Vague instruction: The user's prompt is too ambiguous or underspecified for a single, correct function call,
    yet the benchmark expects one.

### Environment

This category covers flaws within the agent's operating environment---the tools and API results---which can
    make a task unsolvable regardless of the agent's logic.

* Flawed function response: The pre-computed API response provided in the benchmark is incorrect, misleading,
    or doesn't contain the information needed to fulfill the user's request. Since the agent relies on these
    responses, a flawed API response makes the task unsolvable.
  * Look for:
    * Incorrect resolution: An ambiguous name in the function call is resolved to the wrong entity in the
        response.
    * Irrelevant results: The API returns a list of items that are completely irrelevant to the user's request

* Insufficient toolsets: the environment does not provide the necessary tools (functions), making the agent
    impossible to solve the task even with a combination of multiple tools and reasoning.

* Flawed function design: the naming or the description of an available function is misleading or contradicts
    its actual functionality.

### Ground-Truth

This category addresses errors in the provided ground-truth trajectory, where the supposed correct solution is
    itself incorrect, forcing any correct agent to fail the evaluation.

* Malformed function calls: A technical error where a ground-truth function call violates the provided API
    schema.

* Incorrect function calls: A function call is syntactically valid but logically flawed. The function choice
    or a parameter value contradicts the user's request or the context from previous steps.
  * Unjustified/Hallucinated Parameters: A value (e.g., a date, a coordinate) that appears without any
      grounding context.
  * Contradictory Parameter Values: A value that directly contradicts a constraint in the user's prompt.
  * Misspelled or Incorrectly Identified Parameter Values: A misspelled name or an ID/slug that points to the
      wrong entity.

* Redundant/ungrounded function calls: The ground truth function call trajectory consists of function calls
    that are redundant in solving the task, ungrounded by the context, or irrelevant in solving the task.
  * Irrelevant tool call: A function call in the ground truth trajectory is totally irrelevant to the task or
      belongs to a completely different domain.
  * Redundant tool call: A function call that is not necessary in solving the task.
```

```
## Evaluation and Output Format
Carefully analyze the provided sample. Think step-by-step to determine if the ground-truth trajectory is a
     correct and logical solution to the user's prompt.

Your final output must be a JSON object with the following structure, with no additional commentary:

'''json
{{
  "reasoning": "Provide a clear, step-by-step explanation for your decision. If the ground-truth is flawed,
       specify which argument is incorrect and why it contradicts the prompt or schema. If it is not flawed,
       briefly explain why the ground-truth is a correct interpretation of the user's request.",
  "reasoning_summary": "A shorter rationale for your decision. If the ground-truth is not flawed, just mention
       that it is not flawed. If the ground-truth is flawed, specify the issue concisely.",
  "error_category": "The category that corresponds to the issue. e.g., \"Flawed function response\". If the
       sample is not flawed, use \"Not Flawed\".",
  "is_flawed": <true_or_false>

}}

## Sample to be evaluated

### User's Prompt

'''
{instruction}
'''

### List of available functions and their schema

'''json
{available_function_list}
'''

### Ground-truth function call trajectory
* Note that messages with "role": "observation" are the results of the function call right before.

'''json
{gt_conv_traj}
'''
```

## D.2. Generic Evaluation Prompt Template

To compare the effectiveness of our proposed taxonomy-guided prompt, we utilize a generic critique prompt with the same benchmark components and context, but without guidance using our component-based issue-taxonomy.

**Generic Evaluation Prompt for CFB**

```
You are an expert evaluator for ComplexFuncBench, a benchmark designed to assess the complex function-calling
     capabilities of LLMs.
Your task is to review a sample in the benchmark and decide if it has a flaw, which would make it an
     unreliable sample for evaluation.

You will be provided with:

* **User Prompt:** The user's task description and instructions.
* **Available Functions:** A complete list of functions the agent can use, including their JSON schemas.
* **Ground-Truth Trajectory:** The reference sequence of function calls. Note that messages with '"role":
     "observation"' are the outputs of the immediately preceding function call.

## Evaluation and Output Format
Carefully analyze the provided sample. Think step-by-step to determine if the sample has a flaw.

Your final output must be a JSON object with the following structure, with no additional commentary:

'''json
{{
  "reasoning": "Provide a clear, step-by-step explanation for your decision.",
  "is_flawed": <true_or_false>

}}
```

```
```
## Sample to be evaluated

### User's Prompt

```
{instruction}
```

### List of available functions and their schema

```json
{available_function_list}
```

### Ground-truth function call trajectory
* Note that messages with "role": "observation" are the results of the function call right before.

```json
{gt_conv_traj}
```
```

## D.3. Ablation: Effect of the Adversarial Rebuttal Stage

We evaluate the effect of the adversarial rebuttal stage as an auxiliary refinement to the taxonomy-guided LLM judgment. This analysis focuses on ACEBench and CFB, using the same human-validated labels as in prior sections.

Across both benchmarks, the rebuttal consistently improves precision, reducing false positives among flagged issues. On ACEBench, precision increases from 0.865 to 0.918, and on CFB from 0.878 to 0.910. This improvement comes with a moderate reduction in recall, reflecting a trade-off between conservativeness and coverage.

As the primary taxonomy-guided judge already achieves strong alignment with expert annotations, we treat rebuttal as an optional refinement to improve precision and facilitate downstream human verification by reducing cognitive load with the rebuttal reasoning trace, rather than as a required component of the pipeline.

We also report the prompt used in the adversarial rebuttal stage that is provided along with the original judgement and benchmark content.

---

**Taxonomy-guided LLM Evaluation Prompt for CFB**

```
You previously judged benchmark sample '{question_id}' in benchmark '{benchmark}' to be flawed.

Review your previous assessment where you flagged the Ground Truth (GT) trajectory as flawed (Issue=True).

Your goal is to determine if this verdict was a **False Positive** or a **True Positive**.

Apply the following three adjudication principles to specific failure types:

### **1. The "Workaround" Principle (vs. 'Insufficient Toolset' & 'Redundant Tools')**

* **Guideline:** Agents often use "implicit workarounds" when a direct tool is missing.

* **Verdict:** Do not flag a trajectory as flawed if the GT uses existing tools to **logically satisfy the task**, even if it is inefficient.

### **2. The "Semantic Equivalence" Principle When the Benchmark Evaluation System Considers Semantics (vs. 'Incorrect Tool-Call Response')**

* **Guideline:** Some benchmarks consider fuzzy matching (semantic relevance) for evaluation, not just rigid string matching.

* **Verdict:** For these benchmarks, tool usage (e.g., parameters) and tool call results can still be valid without exact string matching.

### **3. The "Derivability" Principle (vs. 'Ungrounded/Hallucinated Parameters')**

* **Guideline:** The GT trajectory often does not include the entire action trajectory. **Searchable data** (valid implicit steps) that can be obtained with the provided tools and task/user context is valid.
```

```
On the other hand, condition checks and parameter data that is impossible to derive from the provided toolset
    and task/user context would indicate a truly flawed task.

**Response Task:**

Based on these guidelines, review the provided benchmark content and your Original Reasoning.

* If the original reasoning violates these principles (is too strict on workarounds/data), **Overturn** the
    judgment (Issue=False).

* If the reasoning holds up (agent failed a logic condition or used truly impossible data), **Maintain** the
    judgment (Issue=True).

Respond using the same JSON schema as your previous reply.
```

## D.4. Ablation Across LLM Judges

We evaluate whether COBA's performance depends on the specific LLM judge by comparing Gemini-2.5-Pro-Thinking, Claude-4-Opus-Thinking, and DeepSeek-V3.1-Thinking across benchmarks. As shown in Table 9, all three judges exhibit consistent alignment with human annotations, although they differ in precision–recall trade-offs. Gemini-2.5-Pro-Thinking achieves average precision/recall of 0.862/0.787, Claude-4-Opus-Thinking achieves 0.861/0.624, and DeepSeek-V3.1-Thinking achieves 0.748/0.846. These results indicate that the taxonomy-guided evaluation framework is not specific to a single judge model, and that lower-cost alternatives can be used in practice depending on the desired precision–recall trade-off.

*Table 9.* Precision and recall metrics for Gemini-2.5-Pro-Thinking, Claude-4-Opus-Thinking, and DeepSeek-V3.1-Thinking across benchmarks.

| Benchmark | Gemini-2.5-Pro-Thinking | | Claude-4-Opus-Thinking | | DeepSeek-V3.1-Thinking | |
|---|---|---|---|---|---|---|
| | **Prec** | **Rec** | **Prec** | **Rec** | **Prec** | **Rec** |
| ACEBench | 0.865 | 0.882 | 0.778 | 0.778 | 0.781 | 0.694 |
| BFCL V3 | 0.865 | 0.800 | 0.917 | 0.550 | 0.745 | 0.950 |
| CFB | 0.878 | 0.720 | 0.825 | 0.660 | 0.742 | 0.920 |
| $\tau$-Bench | 0.846 | 0.733 | 1.000 | 0.400 | 0.750 | 0.800 |
| $\tau^2$-Bench | 0.857 | 0.800 | 0.786 | 0.733 | 0.722 | 0.867 |
| DrafterBench[1] | - | - | - | - | - | - |

*Notes.* [1] DrafterBench: all issues detected by rule-based filtering.

## D.5. Failure Analysis of LLM-Based Judgments

We analyze cases where the taxonomy-guided LLM judgment disagrees with human annotations to characterize residual errors. As shown in Table 10, false positives mainly arise from overly strict reasoning under partial observability, where the judge penalizes valid but non-canonical solutions or assumes that the provided trajectory is complete. False negatives are dominated by missed low-level specification violations, especially malformed tool calls. Overall, these errors concentrate in a small number of recurring patterns, suggesting clear directions for targeted improvement, such as stronger schema-level checks and prompts that more explicitly account for partial trajectories.

*Table 10.* Breakdown of false positive (FP) and false negative (FN) evaluation errors by taxonomy category.

| Type | Category | % |
|---|---|---|
| FP | Partial trajectory | 17.7 |
| FP | Logical inference failure | 11.3 |
| FP | Redundancy misjudgment | 8.1 |
| FP | Insufficient context | 1.6 |
| FN | Malformed tool-calls | 46.8 |
| FN | Incorrect tool responses | 8.1 |
| FN | Insufficient toolset | 4.8 |
| FN | Ambiguity overlooked | 1.6 |

# E. Full Leaderboard of ACEBench

In this appendix, we provide the full leaderboard for ACEBench in Table 11. 18 out of 30 models have their rankings shifted after cleaning.

*Table 11.* Model ranking changes compared for ACEBench for initial and issue-cleaned benchmarks. Dark green indicates rank improvement, and dark red indicates a drop in rank.

| Model | Initial Perf (Rank) | Cleaned Perf (Rank) |
|---|---|---|
| gpt-4o-20240806 | 85.2 (1) | 88.3 (1) |
| gpt-4.1 | 84.2 (2) | 87.2 (2) |
| gpt-4.1-mini | 82.0 (3) | 84.7 (4) |
| DeepSeek-V3.2-Exp-thinking-off | 81.6 (4) | 84.5 (5) |
| claude-4-sonnet-thinking-off | 81.3 (5) | 84.5 (6) |
| claude-4.5-sonnet-thinking-on-10k | 81.1 (6) | 85.0 (3) |
| claude-4-opus-thinking-off | 81.0 (7) | 84.5 (7) |
| Kimi-K2-Instruct-0905 | 80.8 (8) | 83.9 (12) |
| Kimi-K2-Instruct | 80.8 (9) | 83.6 (13) |
| o3-high | 80.6 (10) | 84.1 (8) |
| gpt-oss-120b | 80.4 (11) | 83.1 (16) |
| Gemini-2.5-flash-thinking-off | 80.3 (12) | 84.0 (11) |
| DeepSeek-V3-0324 | 80.0 (13) | 83.2 (14) |
| Gemini-2.5-pro-thinking-on | 79.9 (14) | 84.0 (9) |
| claude-4-opus-thinking-on-10k | 79.7 (15) | 84.0 (10) |
| DeepSeek-V3.2-Exp-thinking-on | 79.5 (16) | 82.9 (17) |
| Qwen3-235B-A22B-Instruct-2507-FP8 | 79.2 (17) | 82.6 (19) |
| DeepSeek-R1-0528 | 78.9 (18) | 82.9 (18) |
| Gemini-2.5-flash-thinking-on | 78.8 (19) | 83.1 (15) |
| gpt-4o-mini | 78.0 (20) | 81.2 (22) |
| Qwen3-235B-A22B-FP8 | 77.7 (21) | 81.6 (20) |
| Claude-4-sonnet-thinking-on-10k | 77.7 (22) | 81.4 (21) |
| gpt-oss-20b | 77.1 (23) | 80.8 (23) |
| Qwen3-235B-A22B-Thinking-2507-FP8 | 76.1 (24) | 80.7 (24) |
| Qwen3-Coder-480B-A35B-Instruct-FP8 | 75.2 (25) | 77.3 (25) |
| o4-mini-high | 73.4 (26) | 77.1 (26) |
| gpt-5 | 71.1 (27) | 76.4 (27) |
| gpt-4.1-nano | 59.0 (28) | 62.7 (28) |
| claude-4.5-sonnet-thinking-off | 56.5 (29) | 58.4 (29) |
| gpt-5-nano | 48.1 (30) | 50.2 (30) |

