# OpenReview forum: "AgentSuite: Toward More Reliable Agent Evaluation with a Component-Based Benchmark Auditing Pipeline"
_ICML.cc/2026/Conference — ICML 2026 regular_

### Official Review · Reviewer_3QT4 · 2026-02-19

**Soundness:** 2
**Presentation:** 3
**Significance:** 2
**Originality:** 3
**Overall Recommendation:** 4
**Confidence:** 4

**Summary:**

First and foremost, the paper proposes a tool for evaluating the quality of agent benchmarks (COBA). This tool revolves around a component-based taxonomy for diagnosing validity issues in agent benchmarks, which the authors also claim as a contribution. Next, the authors use their tool to evaluate several SOTA agent benchmarks, comparing the outputs of their tool to ground truth labels by (alleged) human experts. Last but not least, they authors claim to be releasing "cleaned benchmark artifacts and design guidelines to support more reliable future benchmarks."

**Compliance With Llm Reviewing Policy:**

Affirmed.

**Final Justification:**

Appreciate the conversation. I altered my score based on the rebuttal and did not hear back further. This will be my final score.

**Key Questions For Authors:**

-what are the details of your human expert annotation? who were they? how many of them were there? how many did each annotate? what were the instructions given to them?
-why did you choose to evaluate the six models you chose to evaluate? why not more?
-what are the limitations on your work? this is seriously under-addressed.
-what quality issues might you have ignored?
-where are the cleaned artifact and design guidelines contributions you claim in the Intro? make this clearer.

**Limitations:**

no, definitely not. authors should make a good faith effort to:
-discuss your reliance on LLMs in the pipeline, and the risks of that
-discuss the fact that some quality issues may not be covered here due to the potentially arbitrary way you collected them
-discuss any shortcomings in your human expert evaluation (after disclosing details)
-discuss limitations due to the arguably small number of agent benchmarks evaluated using the pipeline (and test for generalization)

**Strengths And Weaknesses:**

strengths:
-paper is well-written and clear
-illustrations are clear
-the authors make a clear and compelling case for the auditing of agent benchmarks: the problem seems to hold water
-while I'm not totally convinced of its objectivity/comprehensiveness or that it's a major contribution on its own, the taxonomy is certainly interesting and valuable
-the pipeline seems to address the quality issues that are targeted by the author (whether these issues are comprehensive is another issue, discussed below)
-though it has some flaws (discussed below), the evaluation is sound from a design perspective
-the analysis of the benchmark shortcomings is interesting and potentially quite valuable

weaknesses:
-human experts: the evaluation of the method depends entirely on human experts, and you never once explain who these experts are, how many of them there were, or how they were instructed. I don't believe the paper can be accepted at the conference without this information. You should include details on the criteria for selecting these human experts, probably in the main body of the paper, and include the annotation instructions as an appendix. You should also specify how disagreements were negotiated. If these issues are not addressed, unfortunately I fear it materially undermines the significance of the work.
-reproducibility: I'm not clear on whether you have released the "cleaned benchmark artifacts and design guidelines to support more reliable future benchmarks." Are they in the appendix? There is no link to a repo or materials attached to the submission, as far as I can tell. If there is a repo with this material, please put the link in the abstract.
-scope of evaluation: I think there needs to be a better explanation, probably in 2.1, for why you only chose to evaluate the framework on six agent benchmarks (along with the two held out to test generalization). There are other benchmarks in the literature, as you point out in your related work section (you do overlook some, like: https://arxiv.org/html/2508.16260v2). Why focus on just six, if the framework is easily scaled to new agent benchmarks? And why only use two to test generalization? If these numbers are kept, the danger of overfitting should be addressed in a limitations section.
-comprehensiveness of quality issues targeted: 2.2 attempts to leverage the component taxonomy to "turn[] scattered anecdotes into actionable categories and directly inform[] the modular detectors used in our pipeline"....however, these author-chosen actionable categories wind up seeming a bit arbitrary and subjective; we can't be convinced of their comprehensiveness. Can the authors do anything to ensure that these actionable categories cover all quality issues that might arise for each component...and make them more objective-feeling? Would a systematic survey of issues with existing benchmarks, perhaps by human expert annotators, ensure comprehensiveness? If comprehensiveness is not achieved, the tool could provide false hope in benchmarks that suffer from quality issues that are not covered here.
-reliance on LLMs: the pipeline relies heavily on LLMs and, while the authors take some precautions, ultimately inherits any inaccuracies on their part. This should be discussed in more detail.
-limitations section: the limitations section feels very perfunctory and does not address some of the issues above - or other limitations - in a substantive way. More than half of this section is simply spent recapping earlier parts of the paper. It is inadequate.
-nit pick: weird comma issues in first paragraph of Intro

---

> ### Author Rebuttal · Authors · 2026-03-31
>
> Thank you for your time and constructive suggestions! We have addressed all your comments below and will revise the limitations section to explicitly reflect these discussions.
>
> ---
> **Q1 Human Annotation Protocol and Expert Agreement**
>
> We employed 5 annotators, graduate researchers with prior publications or research experience in LLMs and agent systems, who were trained on annotation guidelines regarding benchmark specifications and implementation details. For each benchmark, 3 annotators independently labeled each task for (i) validity (e.g., solvability, ambiguity, ground-truth correctness, evaluation fairness, cross-component consistency) and (ii) issue category using our taxonomy, with definitions and representative benchmark examples provided.
>
> We observe strong agreement (Cohen’s $\kappa$ = 0.824). Disagreements, primarily borderline cases involving ambiguous instructions or underspecified ground truth, were resolved via discussion grounded in benchmark design intent and evaluation logic. We will include relevant details on annotators, the annotation process, and agreement metrics in the revision to improve transparency and reproducibility.
>
> ---
> **Q2 Details on taxonomy derivation and coverage**
>
> The taxonomy is constructed via a bottom-up, data-driven process with 5 annotators. For each benchmark, 3 annotators independently reviewed sampled tasks and trajectories without a predefined schema, proposed issue types based on recurring patterns, and iteratively refined categories through comparison and expanded sampling by merging overlaps, splitting ambiguities, and resolving disagreement cases. We progressively expanded coverage until categories stabilized, resulting in 21–29% task coverage (out of total 5086 tasks) across benchmarks spanning diverse task types and difficulty levels.
>
> We clarify that we do not claim complete exhaustiveness, as a full manual audit was not performed and new issue types may arise. Instead, the taxonomy captures recurring patterns and is extensible to incorporate new categories. We will clarify this scope in the revision.
>
> ---
> **Q3 Benchmark Coverage, Generalization, and Scaling**
>
> We clarify that the six benchmarks were selected to cover representative domains and widely used interactive, tool-based agent settings, rather than maximize count. They span diverse designs used in both industry and academic evaluations, allowing us to validate the pipeline across heterogeneous benchmarks.
>
> “Minimal adaptation” for generalization refers to a lightweight parsing step that maps each benchmark’s design and implementation (e.g., instructions, tools, ground truth, evaluation rules) to a unified standardized format. This step is primarily structural, and the pipeline itself remains unchanged. It enables human experts to focus on verifying flagged issues to establish reliable ground truth. These verification efforts, rather than pipeline adaptation, are heavily required to fully audit the benchmarks, which remains a limitation of this research.
>
> To further demonstrate generalization, we apply COBA to WebArena (Zhou et al., 2023), a structurally different web-based benchmark. On a stratified subset of 20 tasks, we identify 3 validated issues (e.g., ground-truth inconsistencies, malformed task configurations) consistent with our taxonomy. While preliminary, this result provides additional evidence that the pipeline adapts to new benchmarks without redesign.
>
> ---
> **Q4 Reliance on LLM-Based Judgment and Mitigation**
>
> We clarify that LLM-based judgment is used only for initial issue detection, and these candidate flags may inherit limitations of LLM-based judgment. COBA explicitly separates detection from final validation: the LLM judge flags candidate issues and provides structured reasoning, while all reported issues and labels are human-verified prior to being finalized. This ensures that final evaluations reported reflect validated benchmark flaws rather than artifacts of LLM judgment. This design preserves scalability by narrowing the search space and reducing human effort, while maintaining reliability through human verification.
>
> To further assess the impact of LLM-based judgment, we evaluate multiple judge models (Gemini-2.5-Pro, Claude-4-Opus, and DeepSeek-V3.1) on all benchmarks. All show consistent alignment with human annotations as shown below, indicating that conclusions are robust to the choice of LLM judge.
> |Model|Avg. Prec.|Avg. Rec.|
> |:-|:-:|:-:|
> |Gemini-2.5|0.862|0.787|
> |Claude-4|0.861|0.624|
> |DS-V3.1|0.748|0.846|
>
> ---
> **Q5 Reproducibility and Cleaned Benchmark Artifacts**
>
> We include the anonymized repository for the cleaned benchmark artifacts for reproducibility and community use (Repo: https://anonymous.4open.science/r/AgentEvalSuite-F658/README.md). The cleaned artifacts include human-verified fixes guided by our taxonomy and pipeline (extending App A.3), fixing 86% of identified issues.

---

> > ### Author Rebuttal · Reviewer_3QT4 · 2026-04-01
> >
> > I am satisfied with the response and will adjust my score accordingly, but kindly ask the authors to update the paper, if accepted, with the details that that they have provided to me in this rebuttal (e.g., the repo URL and the description of the annotators).

---

> > > ### Author Response · Authors · 2026-04-08
> > >
> > > We deeply appreciate your constructive feedback throughout the review process, as well as your decision to raise the score. We are glad that our rebuttal has successfully resolved your concerns, and we will incorporate the discussed updates and clarifications in the final manuscript.

---

### Official Review · Reviewer_gW8j · 2026-03-03

**Soundness:** 3
**Presentation:** 3
**Significance:** 2
**Originality:** 3
**Overall Recommendation:** 4
**Confidence:** 4

**Summary:**

The paper presents an automated pipeline to detect issues in agentic benchmarks. The pipeline aims to identify issues across four benchmark components, namely, User, Environment, Ground Truth and Evaluation. For each component, a taxonomy of issues is derived, and rule-based detectors and taxonomy-guided LLM evaluation is used to identify them. Experimental results show strong alignment between expert and LLM evaluation. The findings show that existing benchmarks have several flaws and the pipeline can help identify issues across benchmarks.

**Compliance With Llm Reviewing Policy:**

Affirmed.

**Final Justification:**

The rebuttal answered my questions, such as round the annotation protocol. Additional analyses like LLM-as-judge robustness increases the soundness of the work.

**Key Questions For Authors:**

1. Are the results in Table 6 reported over multiple runs? What is the std. deviation?
2. What is the human annotation protocol? How many annotators were involved and what is the inter-annotator agreement?

**Limitations:**

yes

**Strengths And Weaknesses:**

Strengths:
1. The paper tackles an important problem. With the increase in agentic benchmarks, there is a need to automatically ensure their quality.
2. The paper provides a systematic and useful breakdown of issues in agentic benchmarks across various components.
3. Experimental results show a strong alignment of expert judgement with LLM evaluation and identify major flaws in existing benchmarks.

Weaknesses:
1. Minimal details on how the taxonomy was derived. How were the issues per component derived initially? Were multiple annotators involved in the process? What is the "lightweight mapping" mentioned in Section 3?
2. Only a single judge model, gemini-2.5-pro is used, which limits the generalizability of the framework.
3. Table 4 aims to show the generalizability of the framework. However, it does not mention if those benchmarks could have other issues that are not covered by the current taxonomy.
4. To me, the taxonomy is the main contribution of the paper. Otherwise, it is mainly LLM-based benchmark evaluation. However, for the ablation in Section 4.4 samples only 20% of 316 cases (63) for evaluation, which is a very small number to draw meaningful conclusions.

---

> ### Author Rebuttal · Authors · 2026-03-31
>
> We thank the reviewer for the insightful comments and questions. We have addressed all your comments and suggestions as follows.
>
> ---
> **Q1 Human Annotation Protocol and Expert Agreement**
>
> We employed 5 annotators, graduate researchers with prior publications or research experience in LLMs and agent systems, who were trained on annotation guidelines regarding benchmark specifications and implementation details. For each benchmark, 3 annotators independently labeled each task for (i) validity (e.g., solvability, ambiguity, ground-truth correctness, evaluation fairness, cross-component consistency) and (ii) issue category using our taxonomy, with definitions and representative benchmark examples provided.
>
> We observe strong agreement (Cohen’s $\kappa$ = 0.824). Disagreements, primarily borderline cases involving ambiguous instructions or underspecified ground truth, were resolved via discussion grounded in benchmark design intent and evaluation logic. We will include relevant details on annotators, the annotation process, and agreement metrics in the revision to improve transparency and reproducibility.
>
> ---
> **Q2 Details on taxonomy derivation and coverage**
>
> The taxonomy is constructed via a bottom-up, data-driven process with 5 annotators. For each benchmark, 3 annotators independently reviewed sampled tasks and trajectories without a predefined schema, proposed issue types based on recurring patterns, and iteratively refined categories through comparison and expanded sampling by merging overlaps, splitting ambiguities, and resolving disagreement cases. We progressively expanded coverage until categories stabilized, resulting in 21–29% task coverage (out of total 5086 tasks) across benchmarks spanning diverse task types and difficulty levels.
>
> We clarify that we do not claim complete exhaustiveness, as a full manual audit was not performed and new issue types may arise. Instead, the taxonomy captures recurring patterns and is extensible to incorporate new categories. We will clarify this scope in the revision.
>
> ---
> **Q3 Expanded Validation of Taxonomy Effectiveness (Addressing Sample Size)**
>
> We have extended the ablation (Sec 4.4) in two ways: (i) we evaluate all disagreement cases from the original benchmarks (316 total), and (ii) we include an additional benchmark (BFCL v3) with 373 disagreement cases, from which we sample 60% (224 cases).
> Across these expanded settings, the taxonomy-guided prompt consistently outperforms a generic critique prompt as shown below.
> |Bench.|Disag.|Win%|
> |:---|:---:|:---:|
> |ACEBench|114|72.8%|
> |CFB|202|73.3%|
> |BFCLv3|373|84.8%|
>
> ---
> **Q4 Robustness to Choice of LLM Judge**
>
> We test robustness across judge models (Gemini-2.5-Pro, Claude-4, DeepSeek-V3.1) on all benchmarks, all showing consistent alignment with human labels (avg precision/recall: 0.862/0.787, 0.861/0.624, 0.748/0.846), indicating results are not model-specific. This also suggests that lower-cost alternatives can be used in practice without significant degradation.
>
> |Benchmark|Metric|Gemini-2.5|Claude-4|DeepSeek-V3.1|
> |:-|:-|:-:|:-:|:-:|
> |ACEBench|Prec.|0.865|0.778|0.781|
> | |Rec.|0.882|0.778|0.694|
> |BFCL V3|Prec.|0.865|0.917|0.745|
> | |Rec.|0.800|0.550|0.950|
> |CFB|Prec.|0.878|0.825|0.742|
> | |Rec.|0.720|0.660|0.920|
> |$\tau$-bench|Prec.|0.846|1.000|0.750|
> | |Rec.|0.733|0.400|0.800|
> |$\tau^2$-bench|Prec.|0.857|0.786|0.722|
> | |Rec.|0.800|0.733|0.867|
>
> *\*DrafterBench: all issues detected by rule-based filtering.*
>
> ---
> **Q5 Statistical Validation of Table 6 Rankings**
>
> Table 6 is based on a single evaluation run per model due to computational cost, and we therefore do not report run-to-run standard deviations. Instead, to assess whether ranking changes induced by issue filtering are meaningful, we compare against a controlled baseline. Specifically, we remove the same number of tasks at random (100 trials) and measure ranking changes relative to the original ranking using Kendall’s $\tau$ and pairwise flip rates.
>
> Our cleaned benchmark produces substantially larger ranking shifts than random removal:
> |Metric|Clean|Rand [95% CI]|
> |:-|:-:|:-:|
> |$\tau$|0.871|0.960 [0.935, 0.981]|
> |Flip (%)|5.75|1.50 [0.46, 2.65]|
>
> All observed values lie outside the 95% confidence intervals of the random baseline, indicating that the ranking changes are not explained by random perturbations, but reflect systematic effects from filtering benchmark issues.
>
> ---
> **Q6 “Lightweight mapping” Clarification**
>
> We clarify that “lightweight mapping” refers to a simple data-loading step. For each benchmark, we implement a loader that extracts its native fields (e.g., instructions, tool schemas, ground truth, evaluation rules) and maps them into our standardized representation $T = \langle U, E, G, V \rangle$. This step is primarily structural, and the LLM-judge prompt follows a consistent structure, with minor benchmark-specific context (e.g., evaluation rules or design intent) included for correct interpretation.

---

> > ### Author Rebuttal · Reviewer_gW8j · 2026-04-02
> >
> > Thank you for the detailed response and additional analyses! Please update the paper accordingly. I have revised the score

---

> > > ### Author Response · Authors · 2026-04-08
> > >
> > > Thank you for your thoughtful and constructive feedback during the review process, as well as for updating your score. We are glad that our rebuttal fully addressed your concerns, and we will make sure to reflect the corresponding clarifications and updates in the final manuscript.

---

### Official Review · Reviewer_xVzc · 2026-03-11

**Soundness:** 2
**Presentation:** 2
**Significance:** 2
**Originality:** 3
**Overall Recommendation:** 3
**Confidence:** 3

**Summary:**

This paper addresses the problem of how to reliably evaluate the performance of LLM-based agents. Existing benchmarks often contain hidden flaws and complex confounders, which can undermine the credibility of leaderboard comparisons. The authors decompose an agent benchmark task into four components User, Environment, Ground Truth, Evaluation), build a component-aligned issue taxonomy, and propose an auditing pipeline combining rule-based detection, taxonomy-guided LLM-as-judge, and an adversarial rebuttal stage.

**Compliance With Llm Reviewing Policy:**

Affirmed.

**Final Justification:**

Thanks to the authors for providing more detailed annotation guidelines. Overall, my assessment of the paper remains borderline. I have raised my score to 3, and I would not oppose its acceptance.

**Key Questions For Authors:**

1. For more complex agent paradigms (e.g., multi-agent systems, long-horizon toolchains), are the four components sufficient? If not, how should the framework be extended or adapted?
2. Does the cleaned benchmark preserve the original capability coverage, or does it change the task distribution in a way that alters what is being measured?

**Limitations:**

Yes.

**Strengths And Weaknesses:**

## Strengths
1. The paper is clearly written, well organized, and easy to follow.
2. By defining a standardized task representation, the approach enables unified handling of heterogeneous agent benchmarks, making cross-benchmark auditing logic more consistent and efficient.

## Weaknesses
1. Using an LLM-as-judge to adjudicate benchmark validity raises reliability concerns. The identified “issues” may partly reflect the judge model’s preferences or phrasing conventions rather than true benchmark flaws.
2. The paper repeatedly cites agreement with “expert judgments,” but provides insufficient details (e.g., number of experts, backgrounds, labeling guidelines, and concrete labeling examples). This weakens the credibility of the claimed alignment with experts.
3. The benchmarks evaluated in the experiments do not appear sufficiently influential or representative. The study would be stronger if it also included widely cited and practically important benchmarks, such as **SWE-bench** .

---

> ### Author Rebuttal · Authors · 2026-03-31
>
> We greatly appreciate the insightful comments and concerns raised, and have addressed them below:
>
> ---
> **Q1 Human Annotation Protocol and Expert Agreement**
>
> We employed 5 annotators, graduate researchers with prior publications or research experience in LLMs and agent systems, who were trained on annotation guidelines regarding benchmark specifications and implementation details. For each benchmark, 3 annotators independently labeled each task for (i) validity (e.g., solvability, ambiguity, ground-truth correctness, evaluation fairness, cross-component consistency) and (ii) issue category using our taxonomy, with definitions and representative benchmark examples provided.
>
> We observe strong agreement (Cohen’s $\kappa$ = 0.824). Disagreements, primarily borderline cases involving ambiguous instructions or underspecified ground truth, were resolved via discussion grounded in benchmark design intent and evaluation logic. We will include relevant details on annotators, the annotation process, and agreement metrics in the revision to improve transparency and reproducibility.
>
> ---
> **Q2 Reliability of LLM-as-Judge**
>
> We clarify that LLM-based judgment is used only for initial issue detection. COBA explicitly separates detection from final validation: the LLM judge flags candidate issues and provides structured reasoning, while all reported issues and labels are human-verified prior to being finalized. This design preserves scalability by narrowing the search space and reducing human effort, while ensuring conclusions reflect validated benchmark flaws rather than model-specific bias.
>
> To further assess the impact of LLM-based judgment, we evaluate multiple judge models (Gemini-2.5-Pro, Claude-4-Opus, and DeepSeek-V3.1) on all benchmarks. All show consistent alignment with human annotations as shown below, indicating that conclusions are robust to the choice of LLM judge.
> |Model|Avg. Prec.|Avg. Rec.|
> |:-|:-:|:-:|
> |Gemini-2.5|0.862|0.787|
> |Claude-4|0.861|0.624|
> |DeepSeek-V3.1|0.748|0.846|
>
> ---
> **Q3 Benchmark Representativeness and Coverage (WebArena Extension)**
>
> We respectfully clarify that our benchmark selection prioritizes widely used, practically relevant interactive, tool-based agent benchmarks spanning diverse settings, including those commonly used in both LLM technical reports and academic work.
>
> While SWE-bench is influential, it focuses on code generation with execution-based evaluation, outside of our scope of interactive, tool-based setting. To broaden coverage, we instead evaluate on WebArena (Zhou et al., 2023), a widely used browser-based agent benchmark aligned with our setting. On a stratified subset of 20 tasks, we identify 3 validated issues (e.g., ground-truth inconsistencies, malformed task configurations) consistent with our taxonomy. While preliminary, these results indicate that the pipeline and taxonomy are applicable to other widely used benchmarks without redesign.
>
> ---
> **Q4 Impact of Filtering on Task Coverage and Distribution**
>
> To assess the distribution of removed tasks across benchmarks, we measure category-level retention. Retention remains high overall (typically ≥0.8) as shown below, with lower-retention concentrated in benchmarks with systematic issues (e.g., role confusion in $\tau$-bench), while others (e.g., DrafterBench, ACEBench subsets) exhibit localized, fixable issues.
>
> We further conduct a follow-up analysis where task-level issues are manually corrected using our taxonomy and pipeline as guidance (extending App A.3; repo: https://anonymous.4open.science/r/AgentEvalSuite-F658/README.md). After applying these fixes, retention improves substantially as shown below.
> |Bench.|Mean Ret. (min)|Fixable (%)|Mean Ret.+Fix (min)|
> |:-|:-:|:-:|:-:|
> |ACEBench|0.84 (0.27)|80%|0.98 (0.86)|
> |CFB|0.85 (0.81)|20%|0.87 (0.84)|
> |$\tau$-bench|0.67 (0.46)|58%|0.86 (0.78)|
> |$\tau^2$-bench|0.70 (0.44)|39%|0.81 (0.62)|
> |BFCLv3|0.84 (0.81)|73%|0.96 (0.95)|
> |DrafterBench|0.33 (0.25)|100%|1.00 (1.00)|
>
> These results show that many removed tasks are recoverable via targeted fixes, as reflected in post-fix retention gains, while remaining low-retention cases reflect systematic benchmark flaws that require broader community effort to address. Overall, filtering removes concentrated invalidity while preserving most capability coverage, as indicated by consistently high retention.
>
> ---
> **Q5 Applying the Taxonomy to Complex Agent Settings**
>
> The four-component decomposition applies to more complex settings (e.g., multi-agent or long-horizon systems). Increased complexity appears within components (e.g., richer interactions, longer trajectories), rather than requiring new top-level categories. When finer granularity is needed, the framework can be extended with sub-components (e.g., per-agent roles or interaction protocols) while preserving the same structure.

---

> > ### Author Rebuttal · Reviewer_xVzc · 2026-04-03
> >
> > I appreciate the response and effort in addressing the review concerns. Since the detailed annotation guidelines are still unclear, and this is fundamental to the reliability of the experimental evaluation, I'm sorry that I do not have sufficient confidence to raise my score.

---

> > > ### Author Response · Authors · 2026-04-06
> > >
> > > We thank the reviewer for the follow-up and for emphasizing that clear annotation guidelines and examples are central to the credibility of our evaluation. We agree that this is a core aspect. Due to space constraints, our prior response compressed these details. We now provide a more structured description of the annotation protocol with explicit guidelines and concrete examples.
> > >
> > > ---
> > > ### **Annotation Setup**
> > >
> > > We employed 5 annotators (graduate researchers with prior experience in LLMs and agent systems). For each task, 3 annotators independently labeled samples before discussion. Annotators were given full access to benchmark artifacts, including task specifications, implementation details (e.g., environment simulation, evaluation system), and 30 model trajectories, enabling an end-to-end understanding for informed judgment. Annotators were onboarded via documentation and code walkthroughs to ensure a shared understanding of the benchmark’s design intent and implementation logic.
> > >
> > > ### **Annotation Protocol**
> > >
> > >
> > > **(1) Annotation Guideline and Calibration**
> > >
> > > The annotation guideline is defined as follows: a task is labeled flawed only when there is clear, observable evidence of a benchmark-intrinsic issue, including (1) inconsistencies across components (user, environment, ground truth, evaluation), (2) missing or incorrect environment support, (3) ground-truth errors or contradictions with the specification, (4) misaligned evaluation criteria, or (5) underspecified instructions preventing well-defined evaluation. Conversely, annotators exclude cases arising from intrinsic difficulty, evaluation-permitted or intentionally introduced ambiguity, and alternative valid solutions consistent with the benchmark design.
> > >
> > > To calibrate decision boundaries and ensure consistent interpretation of the guideline across benchmarks, annotators were provided with representative edge cases and examples. For instance, an ACEBench task is labeled flawed (ground-truth error) when the reference specifies a year (e.g., “2023”) without support in the user prompt, introducing ungrounded information. In contrast, a similar case is labeled valid when “June” is mapped to “2022-06,” as this comes from the tool’s predefined enum mapping (i.e., schema-constrained, not free-form inference). Additional representative examples include cross-component inconsistency (CFB: user requests a taxi in NYC but the ground truth searches in LA) and missing environment support (BFCL-v3: a travel-time request with only a distance tool available), which render tasks inconsistent or unsolvable.
> > >
> > > **(2) Annotation Procedure**
> > >
> > > **(2-a) Validity Judgement**
> > >
> > > Annotators first determine whether a task contains a benchmark-intrinsic validity issue following the annotation guideline, independent of the taxonomy. Each annotator provides a binary label with a rationale grounded in the benchmark specification.
> > >
> > > **(2-b) Issue Categorization**
> > >
> > > For flawed tasks, annotators assign an issue category based on the component where the root cause originates, guided by the taxonomy. For example, a $\tau^2$-bench task attempting to cancel an already departed flight is labeled a Ground Truth error (incorrect tool call), as the reference contradicts the environment state. The taxonomy provides the structured labeling schema used for categorization and is not used to determine validity. It is constructed through a bottom-up, multi-annotator process (similar to prior work such as MAST [1]), and iteratively refined by updating categories, resolving disagreements, and expanding sampled tasks until convergence. This resulted in coverage of ~21–29% of tasks (5086 total) across benchmarks, with all identified issues consistently mappable, indicating strong empirical coverage while remaining extensible.
> > >
> > > **(3) Agreement and Adjudication**
> > >
> > > We observe strong agreement (average pairwise Cohen’s $\kappa$ = 0.824 before discussion), indicating consistent application of the guidelines. Disagreements arise mainly in borderline cases (e.g., underspecification vs acceptable ambiguity). For example, in an ACEBench task where the user requests “some climate data,” annotators initially disagreed on whether to map this to a specific detail-level parameter, with one viewing it as implicit and others as ungrounded. The case was ultimately labeled flawed, as selecting a specific value introduces unsupported information and does not allow for a stable, consistent mapping. All disagreements were resolved through discussion grounded in task specifications and benchmark evaluation criteria. Annotation required substantial effort (e.g., ~13 hours per annotator for 50 $\tau$-bench tasks), reflecting careful joint reasoning over specifications, environment behavior, and evaluation.
> > >
> > > ---
> > > **Reference**
> > >
> > > [1] Why Do Multi-Agent LLM Systems Fail? (Cemri et al., 2025)

---

### Official Review · Reviewer_jxRv · 2026-03-13

**Soundness:** 3
**Presentation:** 3
**Significance:** 3
**Originality:** 2
**Overall Recommendation:** 4
**Confidence:** 4

**Summary:**

* RQ:  How can we systematically detect and filter hidden validity issues and artifacts across the complex, interacting components of large language model (LLM) agent benchmarks?

* Method: COBA (Component-based Benchmark Auditing) - an automated pipeline that decomposes agent tasks into four components (User, Environment, Ground Truth, and Evaluation) and applies hybrid rule-based detectors, taxonomy-guided LLM evaluation, and an adversarial rebuttal stage to find flaws.

* Contribution (COBA)
- aligns strongly with expert judgments (F1 scores of 0.791–0.874)
- uncovers widespread flaws in six widely used benchmarks that manual verification missed
- generalizes well to new benchmarks.

**Compliance With Llm Reviewing Policy:**

Affirmed.

**Key Questions For Authors:**

1. Detail the expert human annotation (who?, how?) for Table 3?
   -> provide cohen / fleiss kappa

2. How does the adversarial rebuttal stage differ mechanically from standard LLM self-reflection or debate, and how does it perform against those baselines?

3. Breakdown COBA's failures (false positives and false negatives)
 - COBA disagrees with human annotators roughly 15-20% of the time.

**Limitations:**

Yes

**Strengths And Weaknesses:**

# Soundness

## Strength
- COBA breaks it into components, making the auditing easier (user, environment, ground truth, eval)
- inclusion of adversarial rebuttal stage to try to mitigate LLM-as-a-judge bias
- Hybrid eval (rule-based & LLM), since rule-based is powerful when known & cheaper, esp when known / security issues


## Weakness
- adversarial rebuttal stage doesn't necessarily mitigate LLM-as-a-judge bias, may lower it
- seems to be generalizable, but the minimal adaption can be an overclaim
- the taxonomny fails to capture temporal / cascading errors (agent benchmarks involve many steps, but step 2 can be wrong, making steps from 2-5 all wrong)
- Cohen K of the expert annotators & more explanation on who these experts were.


# Presentation

## Strength
- strong conceptualization
- reproducible

## Weakness
- adversarial rebuttal stage is very briefly discussed in 3.4. It is sold more in the abstract & conclusion. The ablation study in Appendix D should be brought up
- Where did the pipeline disagree with humans (the ~15-20% error rate)
- Fig 4 can benefit from a better color palette (try to use a more color blind friendly version, as it can be a bit hard to differentiate when one issue starts and ends.

# Significance:

## Strength
- very timely, systematic vulnerability of agent benchmarks
- high practical use

## Weakness
- the main paper focuses "flawed" vs. "not flawed" --> removal of flawed tasks.
   - 35% of a benchmark (as seen in $\mu$-Bench) doesn't seem such a good idea (can skew distribution)
   - authors do look into trying to fix the tasks (Appendix A.3), but the main contribution is throwing away over a constructive auto-correction mechanism, which limits its long-term utility.
- LLM as a judge
- Gemini-2.5-Pro-Thinking (very costlyyy)

# Originality
* creative synthesis

## Weakness
- rule-based system / unit-tests are normal (e.g.,  scripts or regex to check if a ground-truth tool call violates a JSON schema is standard software engineering practice,)
- adversarial rebuttal is really just rebranding of standard LLM self-reflection, critique-and-revise, or multi-agent debate prompting strategies
- providing an LLM with a taxonomy and asking it to find errors is a highly standard application of the LLM-as-a-judge

---

> ### Author Rebuttal · Authors · 2026-03-31
>
> We greatly appreciate the insightful comments, and have addressed them below:
>
> ---
> **Q1 Human Annotation Protocol and Expert Agreement**
>
> We employed 5 annotators, graduate researchers with prior publications or research experience in LLMs and agent systems, who were trained on annotation guidelines regarding benchmark specifications and implementation details. For each benchmark, 3 annotators independently labeled each task for (i) validity (e.g., solvability, ambiguity, ground-truth correctness, evaluation fairness, cross-component consistency) and (ii) issue category using our taxonomy, with definitions and representative benchmark examples provided.
>
> We observe strong agreement (Cohen’s $\kappa$ = 0.824). Disagreements, primarily borderline cases involving ambiguous instructions or underspecified ground truth, were resolved via discussion grounded in benchmark design intent and evaluation logic. We will include relevant details on annotators, the annotation process, and agreement metrics in the revision to improve transparency and reproducibility.
>
> ---
> **Q2 LLM-as-a-Judge Reliability, Failure Breakdown, and Rebuttal**
>
> **Positioning in terms of contribution.**
> While individual stages (rule-based checks, LLM judges, self-critique prompting) are standard, our contribution is their integration through a component-based taxonomy that enables structured, component-localized auditing of interacting benchmark elements, constraining judgment to explicit issues rather than open-ended critique. Consistent with this, our ablation (Sec. 4.4; extension in response to reviewer gW8j) shows taxonomy-guided prompting yields more reliable judgments over generic critique prompting. Performance is also consistent across judge models (results in response to reviewer gW8j), indicating robustness.
>
> **Adversarial rebuttal vs. self-reflection.**
> We clarify that the rebuttal stage is a post-hoc validation step on flagged cases, not iterative refinement of model outputs. It evaluates whether a “flawed” label should be overturned under simple principles (e.g., valid workarounds, parameter derivability, semantic equivalence), operating on evaluation decisions rather than model outputs.
>
> **Pipeline-human disagreement analysis.**
> We analyze disagreement cases to characterize residual errors in the table below. False positives mainly arise from overly strict reasoning under partial observability (e.g., penalizing valid but non-canonical solutions). False negatives are dominated by missed low-level specification violations (e.g., malformed tool calls). These errors concentrate in recurring patterns, indicating predictable failure modes and suggesting clear directions for targeted improvement.
> |Type|Category|%|
> |:-|:-|:-|
> |FP|Partial trajectory|17.7|
> |FP|Logical inference failure|11.3|
> |FP|Redundancy misjudgment|8.1|
> |FP|Insufficient context|1.6|
> |FN|Malformed tool-calls|46.8|
> |FN|Incorrect tool responses|8.1|
> |FN|Insufficient toolset|4.8|
> |FN|Ambiguity overlooked|1.6|
>
> ---
> **Q3 Impact of Filtering on Benchmark Coverage and Repair**
>
> To assess the distribution of removed tasks across benchmarks, we measure category-level retention. Retention remains high overall (typically ≥0.8) as shown below, with lower-retention concentrated in benchmarks with systematic issues (e.g., role confusion in $\tau$-bench), while others (e.g., DrafterBench, ACEBench subsets) exhibit localized, fixable issues.
>
> We further conduct a follow-up analysis where task-level issues are manually corrected using our taxonomy and pipeline as guidance (extending App A.3; repo: https://anonymous.4open.science/r/AgentEvalSuite-F658/README.md). After applying these fixes, retention improves substantially as shown below.
> |Bench.|Mean Ret. (min)|Fixable (%)|Mean Ret.+Fix (min)|
> |:-|:-:|:-:|:-:|
> |ACEBench|0.84 (0.27)|80%|0.98 (0.86)|
> |CFB|0.85 (0.81)|20%|0.87 (0.84)|
> |$\tau$-bench|0.67 (0.46)|58%|0.86 (0.78)|
> |$\tau^2$-bench|0.70 (0.44)|39%|0.81 (0.62)|
> |BFCLv3|0.84 (0.81)|73%|0.96 (0.95)|
> |DrafterBench|0.33 (0.25)|100%|1.00 (1.00)|
>
> These results show that many removed tasks are recoverable via targeted fixes, while remaining low-retention cases reflect systematic benchmark flaws that require broader community effort to address. Overall, filtering removes concentrated invalidity while preserving most capability coverage, as indicated by consistently high retention.
>
> ---
> **Q4 Taxonomy Coverage of Temporal and Cascading Errors**
> These cases are not treated as independent step errors, but as trajectory-level inconsistencies originating from incorrect intermediate actions, which the taxonomy captures as ground-truth correctness violations (i.e., Ground Truth – incorrect tool calls). For example, in BFCLv4 web search tasks, we identified multi-step ground-truth trajectories where an incorrect intermediate action propagates to subsequent steps, resulting in globally flawed trajectories.
>
> ---
> We also thank the reviewer for suggestions on presentation and will revise accordingly.

---

> > ### Author Rebuttal · Reviewer_jxRv · 2026-04-03
> >
> > My scores will be remain.

---

> > > ### Author Response · Authors · 2026-04-08
> > >
> > > We sincerely thank you for your constructive feedback throughout the review process and for maintaining your positive assessment. We are glad that our rebuttal addressed your concerns, and we will incorporate the discussed updates and clarifications in the final manuscript.

---

### Decision · Program_Chairs · 2026-04-30

**Decision:**

Accept (regular)

**Comment:**

Reviewers agreed that as agentic benchmarks proliferate, the need for an automated pipeline to ensure their quality is important and practical. Reviewers expressed some concerns, notably including concerns with the “expert judgments” without details about, e.g., number of experts, backgrounds, labeling guidelines, and concrete labeling examples. Concerns were at least partially resolved during response, e.g., 5 annotators (graduate researchers with prior experience in LLMs and agent systems).



Hallucinations:

Reference: Perez, E. et al. Discovering latent knowledge in language models without supervision. arXiv preprint, 2023. URL https://arxiv.org/abs/2212.03827.
Issue: authors mismatch with arXiv

Reference: Truong, S., Chok, T., Thai, N. H., Ma, Y., and Koyejo, S. Fantastic bugs and where to find them in ai benchmarks. arXiv preprint arXiv:2511.16842, 2025. URL https://arxiv.org/abs/2511.16842.
Issue: authors mismatch with arXiv